# TabReD: Analyzing Pitfalls and Filling the Gaps in Tabular Deep Learning Benchmarks

**Ivan Rubachev**[= 1,2]    **Nikolay Kartashev**[= 2,1]    **Yury Gorishniy**[1]    **Artem Babenko**[1,2]
[1]Yandex    [2]HSE University

## Abstract

Advances in machine learning research drive progress in real-world applications. To ensure this progress, it is important to understand the potential pitfalls on the way from a novel method's success on academic benchmarks to its practical deployment. In this work, we analyze existing tabular deep learning benchmarks and find two common characteristics of tabular data in typical industrial applications that are underrepresented in the datasets usually used for evaluation in the literature. First, in real-world deployment scenarios, distribution of data often changes over time. To account for this distribution drift, time-based train/test splits should be used in evaluation. However, existing academic tabular datasets often lack timestamp metadata to enable such evaluation. Second, a considerable portion of datasets in production settings stem from extensive data acquisition and feature engineering pipelines. This can have an impact on the absolute and relative number of predictive, uninformative, and correlated features compared to academic datasets. In this work, we aim to understand how recent research advances in tabular deep learning transfer to these underrepresented conditions. To this end, we introduce TabReD – a collection of eight industry-grade tabular datasets. We reassess a large number of tabular ML models and techniques on TabReD. We demonstrate that evaluation on both time-based data splits and richer feature sets leads to different methods ranking, compared to evaluation on random splits and smaller number of features, which are common in academic benchmarks. Furthermore, simple MLP-like architectures and GBDT show the best results on the TabReD datasets, while other methods are less effective in the new setting.

## 1 Introduction

During several recent years, research on tabular machine learning has grown rapidly. Plenty of works have proposed neural network architectures (Klambauer et al., 2017; Wang et al., 2020; Gorishniy et al., 2021; 2022; 2024; Chen et al., 2023a;b; Ye et al., 2024) that are competitive or even superior to "shallow" GBDT models (Prokhorenkova et al., 2018; Ke et al., 2017; Chen & Guestrin, 2016), which has strengthened research interest in the field. Furthermore, specialized workshops devoted to table representation learning are organized on the top-tier ML conferences[1], which highlights the importance of this research line to the community.

An important benefit of machine learning research is the practical application of novel methods developed in academia. However, some conditions encountered in real-world deployment can challenge the benefits of the proposed methods. To study how different methods perform under these conditions, one needs a representative group of datasets. In this work, we study the specifics of datasets from the existing academic benchmarks, and which common practical conditions are unrepresented in them. We investigate each dataset from the popular tabular DL benchmarks and identify some of them as synthetic, untraceable or non-tabular. Moreover, we find and highlight eleven datasets containing leaks. Importantly, in our analysis, we found two data characteristics common in industrial tabular ML applications that are underrepresented in the current academic benchmarks.

---

[1]Table Representation Learning Workshop, NeurIPS

First, in practice, data is often subject to a gradual temporal shift. To account for this, in practice, datasets are often split into train/validation/test parts according to timestamps of datapoints (Herman et al., 2024; Stein, 2002; Baranchuk et al., 2023; Ji et al., 2023; Huyen, 2022). This scenario differs from the already covered (Gardner et al., 2023) distribution shift, because the shift is gradual and the time data representing the degree of this shift is available to the developer. In fact, even among academic benchmarks, there is a big number of datasets with strong time dependencies between instances (e.g. electricity market prediction (van Rijn, 2020), flight delay estimation (Ballesteros, 2019), bike sharing demand (van Rijn, 2014), and others). However, even in such cases, random splits are used in papers instead of time-based splits. Which makes it possible to draw conclusions on I.I.D. data, but creates a gap between datasets and a real-world application from which they come. Moreover, timestamp or other task-appropriate split metadata is often simply not available.

Second, we find that datasets with large numbers of predictive features and extensive feature engineering are scarce in academic benchmarks. Conversely, such feature-rich datasets are common in many industrial settings (Fu & Soman, 2021; Simha, 2020; Kakade, 2021; Anil et al., 2022; Wang et al., 2020), but they are often proprietary and unavailable to the academic community.

In light of these discoveries, we study a question of transferability of novel methods in tabular DL to these underrepresented conditions. To answer this question and fill the gap in existing academic benchmarks, we introduce the TabReD benchmark – a collection of eight datasets, all drawn from real-world industrial applications with tabular data. All TabReD datasets come with time-based splits into train, validation and test parts. Furthermore, because of additional investments in data acquisition and feature engineering, all datasets in TabReD have more features. This stems from adopting the preprocessing steps from production ML pipelines and Kaggle competition forums, where extensive data engineering is often highly prioritized.

We evaluate numerous tabular methods on the TabReD benchmark. We find that most of the tested novel architectures and techniques that show improvements on academic benchmarks do not show benefits on our datasets, while simple MLP architecture with embeddings and GBDT methods show top performance across the new benchmark.

To summarize, our paper presents the following contributions:

- We analyze the existing tabular DL benchmarks in academia, find data leakage issues, use of non-tabular, synthetic or untraceable datasets. We find that temporally-evolving and feature-rich datasets are underrepresented in academic benchmarks.

- We introduce TabReD – a collection of eight industry-grade tabular datasets that span a wide range of domains, from finance to food delivery services. With TabReD we increase the coverage of industrial tabular ML use-cases in academic benchmarks.

- We evaluate a large number of tabular DL techniques on TabReD. We find that, in the feature-rich, time-evolving setting facilitated by TabReD: (1) GBDT and MLPs with embeddings (Gorishniy et al., 2022) demonstrate the best average performance; (2) more complex DL methods turn out to be less effective. We demonstrate that correct evaluation on datasets with temporally shifted validation and test sets is crucial as it leads to significant differences in rankings and relative performance of methods, compared to commonly used random-split based evaluation. We also demonstrate that richness of feature sets influences model ranking and negatively affects augmentations and retrieval models. In particular, we observe that XGBoost performance margin diminishes in correct evaluation setups.

## 2 RELATED WORK

**Tabular deep learning** is a dynamically developing research area, with the recent works proposing new model architectures (Klambauer et al., 2017; Wang et al., 2020; Gorishniy et al., 2021; 2022; 2024; Chen et al., 2023a;b), regularizations (Jeffares et al., 2023), (pre-)training pipelines (Bahri et al., 2021; Rubachev et al., 2022; Lee et al., 2024) and other solutions (Hollmann et al., 2023). Since the common way to justify the usage of new approaches is to empirically compare them against the baselines, the choice of the benchmarks for evaluation is critical.

**Tabular deep learning benchmarks**. Since tabular tasks occur in a large number of applications from various domains, there is no single dataset that would be sufficient for evaluation. A typical

tabular DL paper reports the results on several tasks from different domains, usually coming from one of the public data repositories. The two traditional data sources for tabular datasets are the UCI [2] and OpenML[3] repositories, currently holding thousands of datasets. Unfortunately, datasets available in public repositories do not cover all tabular ML use cases. In particular, we find that some conditions of industrial tabular ML applications are underrepresented in these public repositories.

Another source of datasets is the Kaggle[4] platform, which hosts a plethora of ML competitions, including ones with tabular data. Datasets from competitions are often provided by people solving particular real-world problems, making Kaggle an attractive source of datasets for tabular ML research. Surprisingly, many popular benchmarks rely on UCI and OpenML and underutilize tabular datasets from Kaggle. For example, out of 100 academic datasets that we analyze in section 3, only four come from Kaggle competitions. While there are certainly problematic datasets on Kaggle (e.g. containing data leakage, synthetic or duplicate data), with careful selection one can find high quality datasets. To construct TabReD, we utilize several datasets from Kaggle tabular data competitions, and we also introduce four new datasets from real ML production systems at a large tech company.

**Tabzilla** (McElfresh et al., 2023) and the **Grinsztajn et al. (2022) benchmark** have gained adoption in the research community. For example, such papers as Gorishniy et al. (2024), Chen et al. (2023b), Feuer et al. (2024), evaluate performance on these benchmarks. Both benchmarks primarily rely on the OpenML repository as a source of datasets, and filter datasets semi-automatically based on metadata like size and baseline performance. In our work, we look closer at all the datasets and find data-leakage issues and non-tabular, synthetic and anonymous/unknown datasets sometimes "sneaking" through automatic filters. Furthermore, these datasets do not represent conditions of temporal shift and extensive feature engineering that are common in practical applications.

**TableShift** (Gardner et al., 2023) and **WildTab** (Kolesnikov, 2023) propose tabular benchmarks with distribution shifts between train/test subsets. These benchmarks are closer in spirit to TabReD, as both describe evaluation in unrepresented conditions. However, both benchmarks focus on out-of-distribution robustness and provide domain generalization methods comparison. We study a broader set of methods, including recent SoTA tabular neural networks. Furthermore, both benchmarks consider more "extreme" shifts, compared to the more ubiquitous gradual temporal shift which is present in all TabReD datasets.

**The field of benchmarks for time-series** focuses on prediction of target variables in the future, as does our benchmark. Works such as Shchur et al. (2023) and Ansari et al. (2024) both contain benchmarks that focus on data with time-shift. However, TabReD also focuses on feature-rich scenarios, in which time feature does not overwhelm many other features in importance. For an extended discussion of time-series methods and TabReD, see Appendix E.

**Benchmarking Under Temporal Shift**. Wild-Time (Yao et al., 2022) proposed a benchmark consisting of five datasets with temporal shift and identified a 20% performance drop due to temporal shift on average. However, tabular data was not the focus of the Wild-Time benchmark.

The presence of temporal shift and the importance of evaluating models under temporal shifts was discussed in many research and application domains including approximate neighbors search (Baranchuk et al., 2023), recommender systems (Shani & Gunawardana, 2011), finance applications (Stein, 2002; Herman et al., 2024), health insurance (Ji et al., 2023) and in general ML systems best practices (Huyen, 2022).

## 3 A CLOSER LOOK AT THE EXISTING TABULAR DL BENCHMARKS

In this section, we take a closer look at the existing benchmarks for tabular deep learning. We analyze dataset sizes, number of features, the presence of temporal shift and its treatment. We also point out the issues of some datasets that we find notable. The first such issue is the presence of leakage. We believe that an inadvertent usage of datasets with leaks has a negative impact on the quality of evaluation. We provide a list of such datasets in the corresponding section below. The second issue is the untraceable or synthetic nature of the data. Without knowing the source of the data and what it

---

[2] https://archive.ics.uci.edu
[3] https://www.openml.org
[4] https://www.kaggle.com

Table 1: The landscape of existing tabular deep learning benchmarks compared to TabReD. We report median dataset sizes, number of features, the number of datasets with various issues. The "Time-splits" column is reported only for the datasets without issues. We see that the datasets semi-automatically gathered from OpenML (Tabzilla and Grinsztajn et al. (2022)) contain more quality issues. Furthermore, no benchmark besides TabReD focuses on temporal-shift based evaluation and less than half of datasets in each benchmark have timestamp metadata needed for time-based validation availability.

* – the original dataset, introduced in (Malinin et al., 2021) has the canonical OOD split, but the standard IID split commonly used contains time-based leakage.

** – the median full dataset size. In experiments, to reduce compute requirements, we use subsampled versions of the TabReD datasets.

| Benchmark | Dataset Sizes ($Q_{50}$) | | Issues (#Issues / #Datasets) | | | Time-split | | |
|---|---|---|---|---|---|---|---|---|
| | #Samples | #Features | Data-Leakage | Synthetic or Untraceable | Non-Tabular | Needed | Possible | Used |
| Grinsztajn et al. (2022) | 16,679 | 13 | 7 / 44 | 1 / 44 | 7 / 44 | 22 | 5 | |
| Tabzilla (McElfresh et al., 2023) | 3,087 | 23 | 3 / 36 | 6 / 36 | 12 / 36 | 12 | 0 | |
| WildTab (Kolesnikov, 2023) | 546,543 | 10 | 1* / 3 | 1 / 3 | 0 / 3 | 1 | 1 | ✗ |
| TableShift (Gardner et al., 2023) | 840,582 | 23 | 0 / 15 | 0 / 15 | 0 / 15 | 15 | 8 | |
| Gorishniy et al. (2024) | 57,909 | 20 | 1* / 10 | 1 / 10 | 0 / 10 | 7 | 1 | |
| **TabReD** (ours) | 7,163,150** | 261 | ✗ | ✗ | ✗ | ✓ | ✓ | ✓ |

represents, it is unclear how transferable are advances on these datasets. The third issue is the usage of the data that belongs to other domains, e.g. image data flattened into an array of values. While such datasets correspond to a valid and useful task, it is unclear how useful are advances on such datasets in practice, since other domain-specific methods usually perform significantly better for this type of data.

Table 1 summarizes our analysis of 100 unique classification and regression datasets from academic benchmarks (Gorishniy et al., 2022; Grinsztajn et al., 2022; Gorishniy et al., 2024; McElfresh et al., 2023; Chen et al., 2023b; Gardner et al., 2023). We also provide detailed meta-data collected in the process with short descriptions of tasks, original data sources, data quality issues and notes on temporal splits in the Appendix F. Our main findings are as follows.

**Data Leakage, Synthetic and Non-Tabular Datasets**. First, a considerable number of tabular datasets have some form of data leakage (11 out of 100). Leakage stems from data preparation errors, near-duplicate instances or inappropriate data splits used for testing. A few of these leakage issues have been reported in prior literature (Gorishniy et al., 2024), but as there are no common protocols for deprecating datasets in ML (Luccioni et al., 2022), datasets with leakage issues are still used. Here is the list of datasets where we identified leaks: eye_movements, visualizing_soil, Gesture Phase, sulfur, artificial-characters, compass, Bike_Sharing_Demand, electricity, Facebook Comments Volume, SGEMM_GPU_kernel_performance, Shifts Weather (in-domain-subset). For some datasets the data source is untraceable, or the data is known to be synthetic without the generation process details or description – there are 13 such datasets. Last, we find that 25 datasets used in academic benchmarks are not inherently tabular by the categorization proposed in Kohli et al. (2024). These datasets either represent raw data stored in a table form (e.g. flattened images) or homogenous features extracted from some raw data source.

**Dataset Size and Feature Engineering**. We find that most datasets from the academic benchmarks have less than 60 features and less than a hundred thousands instances available. Many academic datasets come from publicly available data, which often contain only high-level statistics (e.g. only the source and destination airport and airline IDs for the task of predicting flight delays in the dataset by Ballesteros (2019)). In contrast, many in-the-wild industrial ML applications utilize as much information and data as possible (e.g. Fu & Soman, 2021; Simha, 2020; Kakade, 2021). Unfortunately, not many such datasets from in-the-wild applications are openly available for research. Kaggle competitions sometimes come close to this kind of industry-grade tabular data, but using competition data is less common in current academic benchmarks (only 4 datasets are from Kaggle competitions). To highlight the difference of previous benchmarks with ours, we provide information on correlated features in subsection A.3.

**Lack of canonical splits or timestamp metadata**. All benchmarks except the ones focused on distribution shift do not discuss the question of data splits used for model evaluation, beyond standard experimental evaluation setups (e.g. random split proportions or cross-validation folds). We find that 53 existing datasets (excluding datasets with issues) potentially contain data drifts related to the passage of time, as the data was collected over time. It is a standard industry practice to use time-based splits for validation in such cases (Shani & Gunawardana, 2011; Stein, 2002; Ji et al., 2023; Huyen, 2022). However, only 15 datasets have timestamps available for such splits.

# 4 Constructing the TabReD benchmark

In this section, we introduce the new **Tab**ular benchmark with **Re**al-world industrial **D**atasets (TabReD). To construct TabReD we utilize datasets from Kaggle competitions and industrial ML applications at a large tech company. We adhere to the following criteria when selecting datasets for TabReD. (1) Datasets should be inherently tabular, as discussed in section 3 and Kohli et al. (2024). (2) Feature engineering and feature collection efforts should be as close as possible to the industry practices. We adapt feature-engineering code by studying competition forums for Kaggle datasets, and we use the exact features from production ML systems for the newly introduced datasets. (3) We also take care to avoid leakage in the newly introduced datasets. (4) Datasets should have timestamps available and should have enough samples for the time-based train/test split.

We summarize the main information about our benchmark in Table 2. The complete description of each included dataset can be found in Appendix B. An analysis of feature collinearity is provided in subsection A.3. We also provide the table with our annotations of Kaggle competition, used to filter datasets from Kaggle in Appendix D.

## 4.1 On the role and limitations of TabReD

We see the TabReD benchmark as an important addition to the landscape of tabular datasets. While the current benchmarks already allow analyzing how well does a method work in I.I.D. conditions with non-rich feature sets, TabReD allows studying interaction of a method with the properties that are underrepresented in current benchmarks, including the time-based splits and the "feature-rich" data representation. And, as we will show in section 5, these properties become a non-trivial challenge for some tabular models and techniques. Thus, TabReD enables an additional evaluation in the industrial-like setting that can help in identifying limitations of novel methods.

However, TabReD is not a replacement for current benchmarks, and has certain limitations. First, it is biased towards industry-relevant ML applications, with large sample sizes, extensive feature engineering and temporal data drift. Second, TabReD does not cover some important domains such as medicine, science, and social data. Finally, we note that the lack of precise feature information may limit some potential future applications of these datasets, like leveraging feature names and descriptions with LLMs.

Table 2: Short description of datasets in TabReD. Numbers in parentheses denote full dataset sizes. We use random subsets of large datasets to make extensive hyperparameter tuning feasible.

| Dataset | # Samples | # Features | Source | Task Description |
|---|---|---|---|---|
| Sberbank Housing | 28K | 392 | Kaggle | Real estate price prediction |
| Ecom Offers | 160K | 119 | Kaggle | Predict whether a user will redeem an offers |
| Homesite Insurance | 260K | 299 | Kaggle | Insurance plan acceptance prediction |
| HomeCredit Default | 381K (1.5M) | 696 | Kaggle | Loan default prediction |
| Cooking Time | 319K (12.8M) | 192 | New | Restaurant order cooking time estimation |
| Delivery ETA | 350K (17.0M) | 223 | New | Grocery delivery courier ETA prediction |
| Maps Routing | 279K (13.6M) | 986 | New | Navigation app ETA from live road-graph features |
| Weather | 423K (16.9M) | 103 | New | Weather prediction (temperature) |

## 5 HOW DO TABULAR DL TECHNIQUES TRANSFER TO TABRED CONDITIONS?

In this section, we use TabReD to analyze how tabular deep learning advances on academic benchmarks transfer to the temporally evolving, feature-rich industrial setting it represents. In subsection 5.1 we introduce our experimental setup and list techniques and baselines considered in our study. In subsection 5.2 we evaluate all these techniques on TabReD and discuss the results. In subsection 5.3 we contrast results on academic datasets with TabReD and show that some techniques useful in academic settings are less effective on TabReD than others. In subsection 5.4 we analyze the influence of temporal shift and rich feature sets on evaluation results and method comparison. We also study distribution shift robustness methods on TabReD in subsection A.2.

### 5.1 EXPERIMENTAL SETUP AND TABULAR DEEP LEARNING TECHNIQUES

**Experimental setup**. We adopt training, evaluation and tuning setup from Gorishniy et al. (2024). We tune hyperparameters for most methods[5] using Optuna from Akiba et al. (2019), for DL models we use the AdamW optimizer and optimize MSE loss or binary cross entropy depending on the dataset. By default, each dataset is temporally split into train, validation and test sets. Each model is selected by the performance on the validation set and evaluated on the test set (both for hyperparameter tuning and early-stopping). Test set results are aggregated over 15 random seeds for all methods, and the standard deviations are taken into account to ensure the differences are statistically significant. We randomly subsample large datasets (Homecredit Default, Cooking Time, Delivery ETA and Weather) to make more extensive hyperparameter tuning feasible. For extended description of our experimental setup including data preprocessing, dataset statistics, statistical testing procedures and exact tuning hyperparameter spaces, see Appendix C. Below, we describe the techniques we evaluate on TabReD.

**Non DL Baselines**. We include three main implementations of Gradient Boosted Decision Trees: XGBoost (Chen & Guestrin, 2016), LightGBM (Ke et al., 2017) and CatBoost (Prokhorenkova et al., 2018), as well-established non-DL baselines for tabular data prediction. We also include Random Forest (Breiman, 2001) and linear model as the basic simple ML baselines to ensure that datasets are non-trivial and are not saturated by simplest baselines.

**Tabular DL Baselines**. We include two baselines from (Gorishniy et al., 2021) – MLP and FT-Transformer. We use MLP as the simplest DL baseline and FT-Transformer as a representative baseline for the attention-based tabular DL models. Attention-based models are often considered state-of-the-art (e.g. Gorishniy et al., 2021; Somepalli et al., 2021; Kossen et al., 2021; Chen et al., 2023a) in tabular deep learning research.

**Other Tabular DL Models**. In addition to baseline methods, we include DCNv2 as it was repeatedly used in real-world production settings as reported by (Wang et al., 2020; Anil et al., 2022). We also test alternative MLP-like backbones in ResNet (Gorishniy et al., 2021) and SNN (Klambauer et al., 2017). We also include Trompt (Chen et al., 2023b), it was shown to outperform Transformer for tabular data variants (Somepalli et al., 2021; Gorishniy et al., 2021) on the benchmark from Grinsztajn et al. (2022). Its strong performance on an established academic benchmark aligns well with our goal of finding out how results obtained on academic benchmarks generalize to TabReD.

**Numerical Feature Embeddings**. We include MLP with embeddings for numerical features from Gorishniy et al. (2022). Numerical embeddings provide considerable performance improvements on academic datasets, and make simple MLP models compete with attention-based models and GBDTs. Furthermore, the success of this architectural modification was replicated on academic benchmarks in Ye et al. (2024) and Holzmüller et al. (2024). We find this simple, effective and proven technique important to evaluate in a new setting.

**Retrieval-based models** are a recent addition to the tabular DL model arsenal. We evaluate TabR from Gorishniy et al. (2024) and ModernNCA from Ye et al. (2024). Both retrieval-based models demonstrate impressive performance on common academic datasets from Gorishniy et al. (2021); Grinsztajn et al. (2022) and outperform strong GBDT baselines with a sizeable margin, which warrants their evaluation on TabReD. We exclude numerical embeddings from both models to test the efficacy of retrieval in a new setting in isolation.

---

[5]Trompt is the only exception, due to the method's time complexity. For Trompt we evaluate the default configuration proposed in the respective paper.

**Improved Training Methodologies** like the use of cut-mix like data augmentations (Somepalli et al., 2021; Chen et al., 2023a) or auxiliary training objectives with augmentations and long training schedules (Rubachev et al., 2022; Lee et al., 2024) have produced considerable gains on academic benchmarks. We include two methods from Rubachev et al. (2022), namely long training with data-augmentations "MLP aug." and a method with an additional reconstruction loss "MLP aug. rec."[6].

**Ensembles** are considered a go-to solution for improving performance in many ML competitions. Ensembles have also shown effectiveness in academic tabular DL settings (Gorishniy et al., 2021; Shwartz-Ziv & Armon, 2021). We test this technique on TabReD as well.

## 5.2 RESULTS

In this section, we evaluate all techniques outlined above. Results are summarized in Table 3. Below, we highlight our key takeaways.

Table 3: Performance comparison of tabular ML models on new datasets. Bold entries represent the best methods on each dataset, with standard deviations over 15 seeds taken into account. The last column contains algorithm rank averaged over all datasets (for details, see the subsection C.2). The ranks in bold correspond to the top-3 classical ML methods and the top-3 DL methods.

| Methods | Classification (ROC AUC ↑) | | | Regression (RMSE ↓) | | | | | Average Rank |
|---|---|---|---|---|---|---|---|---|---|
| | Homesite Insurance | Ecom Offers | HomeCredit Default | Sberbank Housing | Cooking Time | Delivery ETA | Maps Routing | Weather | |
| **Classical ML Baselines** | | | | | | | | | |
| XGBoost | 0.9601 | 0.5763 | **0.8670** | 0.2419 | 0.4823 | 0.5468 | 0.1616 | 1.4671 | **2.9 ± 1.5** |
| LightGBM | 0.9603 | 0.5758 | 0.8664 | 0.2468 | 0.4826 | 0.5468 | 0.1618 | **1.4625** | **3.1 ± 1.5** |
| CatBoost | 0.9606 | 0.5596 | 0.8621 | 0.2482 | 0.4823 | **0.5465** | 0.1619 | 1.4688 | **3.4 ± 1.7** |
| RandomForest | 0.9570 | 0.5764 | 0.8269 | 0.2640 | 0.4884 | 0.5959 | 0.1653 | 1.5838 | 7.8 ± 2.0 |
| Linear | 0.9290 | 0.5665 | 0.8168 | 0.2509 | 0.4882 | 0.5579 | 0.1709 | 1.7679 | 8.8 ± 2.7 |
| **Tabular DL Models** | | | | | | | | | |
| MLP | 0.9500 | 0.6015 | 0.8545 | 0.2508 | 0.4820 | 0.5504 | 0.1622 | 1.5470 | 5.0 ± 1.8 |
| SNN | 0.9492 | 0.5996 | 0.8551 | 0.2858 | 0.4838 | 0.5544 | 0.1651 | 1.5649 | 6.6 ± 2.0 |
| DCNv2 | 0.9392 | 0.5955 | 0.8466 | 0.2770 | 0.4842 | 0.5532 | 0.1672 | 1.5782 | 7.6 ± 2.3 |
| ResNet | 0.9469 | 0.5998 | 0.8493 | 0.2743 | 0.4825 | 0.5527 | 0.1625 | 1.5021 | 5.8 ± 2.0 |
| FT-Transformer | 0.9622 | 0.5775 | 0.8571 | 0.2440 | 0.4820 | 0.5542 | 0.1625 | 1.5104 | 4.8 ± 1.6 |
| MLP (PLR) | 0.9621 | 0.5957 | 0.8568 | 0.2438 | 0.4812 | 0.5527 | 0.1616 | 1.5177 | **3.8 ± 1.4** |
| Trompt | 0.9588 | 0.5803 | 0.8355 | 0.2509 | 0.4809 | 0.5519 | 0.1624 | 1.5187 | 5.4 ± 2.1 |
| **Ensembles** | | | | | | | | | |
| MLP ens. | 0.9503 | **0.6019** | 0.8557 | 0.2447 | 0.4815 | 0.5494 | 0.1620 | 1.5186 | **4.1 ± 2.0** |
| MLP-PLR ens. | **0.9629** | 0.5981 | 0.8585 | **0.2381** | **0.4806** | 0.5518 | **0.1612** | 1.4953 | **2.4 ± 1.5** |
| **Training Methodologies** | | | | | | | | | |
| MLP aug. | 0.9523 | 0.6011 | 0.8449 | 0.2659 | 0.4832 | 0.5532 | 0.1631 | 1.5193 | 6.0 ± 2.0 |
| MLP aug. rec. | 0.9531 | 0.5960 | 0.7453 | 0.2515 | 0.4834 | 0.5541 | 0.1636 | 1.5160 | 6.8 ± 2.8 |
| **Retrieval Augmented Tabulard DL** | | | | | | | | | |
| TabR | 0.9487 | 0.5943 | 0.8501 | 0.2820 | 0.4828 | 0.5514 | 0.1639 | 1.4666 | 6.0 ± 2.2 |
| ModernNCA | 0.9514 | 0.5765 | 0.8531 | 0.2593 | 0.4825 | 0.5498 | 0.1625 | 1.5062 | 5.6 ± 1.6 |

**GBDT and MLP with embeddings (MLP-PLR)** are the overall best models on the TabReD benchmark. These findings suggest that numerical feature embeddings (Gorishniy et al., 2022), which have shown success in academic datasets, maintain their utility in the new evaluation scenario. Ensembles also bring consistent performance improvements to MLP and MLP-PLR in line with the existing knowledge on prior benchmarks and practice.

---

[6]In the original work the methods are called "MLP sup" and "MLP rec-sup", we use subscript aug to highlight that methods use augmentations and differentiate them from traditional supervised baselines

**FT-Transformer** is a runner-up, however, it can be slower to train because of the attention module that causes quadratic scaling of computational complexity w.r.t. the number of features. The latter point is relevant for TabReD, since the TabReD datasets have more features than an average academic dataset.

**SNN, DCNv2, ResNet and Trompt** are no better than the MLP baseline. Although Trompt showed promising results on a benchmark from Grinsztajn et al. (2022), it failed to generalize to TabReD. Furthermore, efficiency-wise, Trompt is significantly slower than MLP, and even slower than FT-Transformer.

**Retrieval-Based Models** prove to be less performant on TabReD. One notable exception is the Weather dataset, where TabR has the second-best result. ModernNCA is closer to the MLP baseline on average, but its benefits seen on academic benchmarks do not transfer to the TabReD setting either. We expand on potential challenges TabReD presents for retrieval-based models in the following section.

**Improved Training Methodologies**. Better training recipes that leverage long pre-training on target datasets mostly do not transfer to the setting presented by TabReD. Homesite Insurance and Weather datasets are the only exceptions, where training recipes bring some performance improvements.

## 5.3 COMPARISON WITH PRIOR BENCHMARKS

To further illustrate the utility of the TabReD benchmark for differentiating tabular DL techniques, we compare results for a range of recently proposed methods on an existing academic benchmark from Gorishniy et al. (2024) and on TabReD. We look at four techniques in this section: improved model architectures (MLP-PLR and XGBoost as a reference classic baseline), ensembling, improved training methodologies ("MLP aug.", "MLP aug. rec.") and retrieval-based models (ModernNCA, TabR). The results are in Figure 1. We summarize our key observations below.

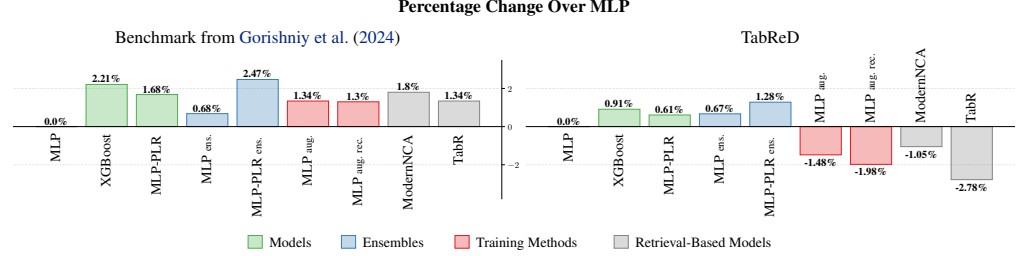

Figure 1: Comparison of tabular DL algorithmic improvements on TabReD and on a popular academic benchmark. We plot average relative percentage improvement over the MLP baseline on both benchmarks. Ensembling and Numerical Embeddings successfully transfer to TabReD. However, success of retrieval-based models and improved training methods is limited to academic benchmarks.

We can see that ensembling and embeddings for numerical features are beneficial on both benchmarks. But two remaining techniques: improved training recipes and retrieval-based models underperform on TabReD, while showing consistently high performance on academic datasets.

We hypothesize that the two TabReD dataset characteristics are at play here. First, feature complexities like multicollinearity and presence of noisy features of more feature-rich TabReD datasets might affect neighbors selection and the feature-shuffle augmentation (for a comparison of feature complexities, see subsection A.3). Second, retrieval-based models rely on an assumption that train objects (used in retrieval) are useful for predicting test instances. This can be violated by the presence of gradual temporal shift. A similar argument may explain the underperformance of long training recipes, as those models might also implicitly "memorize" harder train-set examples through longer training (Rahaman et al., 2019; Feldman & Zhang, 2020). Understanding these phenomena is an interesting avenue for future research.

**Summary**. Overall, the above results reinforce TabReD as an effective tool for uncovering practical failure modes of methods that were previously unidentifiable.

## 5.4 THE EFFECTS OF TEMPORAL SHIFT AND FEATURE ENGINEERING

In this section, we investigate the importance of two TabReD dataset characteristics. For this, we create two variations of the datasets. First, we use random splits instead of the time-based ones. We keep train validation and test set sizes the same and randomly shuffle objects to obtain the random splits. Second, we remove the least important features to imitate setups with less feature engineering, we keep top 20 most important features. We use XGBoost feature importances as the easiest and most performant feature selection method according to Cherepanova et al. (2023).

We are interested in methods ranking and relative performance differences between three dataset variations: default TabReD, randomly-split TabReD and non-"feature-rich" TabReD. For this experiment, we consider multiple techniques: ensembling, MLP aug. rec., MLP-PLR, XGBoost and TabR-S. This selection covers all major paradigms discussed in subsection 5.1 – retrieval-based models, parametric DL, improved training techniques, ensembling and strong non-DL baselines. Furthermore, those methods have diverse performance on our benchmark (see Table 3). Results are summarized in Table 4. An extended discussion and per-dataset results are available in the Appendix A. Below, we highlight key takeaways.

Table 4: Temporal shift and feature engineering influence on model performance. We report the relative percentage improvement to the MLP baseline averaged over all datasets. We see significant performance differences in different conditions. (1) XGBoost performance margin diminishes on temporal data splits. (2) TabR and MLP aug. rec. are better on random splits and datasets with fewer features. (3) MLP-PLR and ensembling are relatively less affected methods and are similarly useful in all three settings.

*∗ – Note that MLP with all features has an advantage of 5.65% over MLP with 20 features.*

|  | MLP ens. | MLP aug. rec. | MLP-PLR | MLP-PLR ens. | TabR | XGBoost |
|---|---|---|---|---|---|---|
| TabReD | 0.67 | -1.98 | 0.61 | 1.28 | -2.78 | 0.91 |
| TabReD *Random splits* | 0.28 | -0.47 | 0.47 | 0.96 | 0.6 | 2.02 |
| TabReD *20 features ∗* | 0.52 | -0.13 | 0.69 | 0.98 | 1.09 | 0.97 |

**Temporal Shift**. We see a significant difference in perfomance of XGBoost and TabR between random and time-based splits. TabR improvement is mostly due to the Weather dataset (see per-dataset results in subsection A.1), while XGBoost improvements are more universal. Interestingly, XGBoost performance margin diminishes on temporal data splits. Ensembling and PLR feature embeddings are stable in terms of relative improvement over MLP on both splits.

**Feature-Rich Datasets**. We see that TabR on datasets with fewer features is among the best performing methods, which may hint to some issues with the L2 distance in neighbors computation or the implicit feature engineering in TabR. Furthermore, improved training methodologies for the MLP (aug. rec.) are less detrimental in this setup, and they are even helpful for some datasets (see per-dataset results in subsection A.3). PLR embeddings and ensembling are stable in terms of relative improvement over MLP on both feature-rich and simplified datasets. Note that feature selection significantly reduces the overall solution quality (by 5.65% on average when comparing MLPs).

**Summary**. From the above results, we conclude that both temporal shifts and rich feature sets are important in TabReD as eliminating these characteristics from datasets can have a significant effect on all aspects involved in the comparison between models: absolute metric values, relative difference in performance and, finally, the relative ranking of models.

## 6 FUTURE WORK

We have demonstrated the importance of taking temporal shifts into account and benchmarked a wide range of prominent tabular DL techniques on TabReD. However, there are still many research questions and techniques like continual learning, gradual temporal shift mitigation methods, missing data imputation and feature selection, that could be explored with TabReD. The question of why some techniques fail to transfer from academic benchmarks to TabReD is worth further investigation, we posit that two key data characteristics in temporal data drift and feature complexities like multicollinearity, noisy features are at play here.

## 7 CONCLUSION

In this work, we analyzed the existing tabular DL benchmarks typically used in the literature and identified their limitations. Also, we described two conditions common in typical deployment scenarios that are underrepresented in current benchmarks. We then composed a new benchmark TabReD that closely reflects these conditions and follows the best industrial practices. We carefully evaluated many recent tabular DL developments in TabReD settings and found that simple baselines like MLP with embeddings and deep ensembles, as well as GBDT methods such as XGBoost, CatBoost and LightGBM work the best, while more complicated tabular DL methods fail to transfer their increased performance from academic benchmarks.

TabReD benchmark facilitates two possible use-cases. First, a method that shows good performance in non-time-shifting and non-feature-rich scenarios set by previous benchmarks could be tested on TabReD, which will demonstrate how a method works under TabReD's conditions. Second, methods specifically focusing on time-shifting data and rich feature sets can be benchmarked on TabReD. We believe that TabReD can serve as an important step towards more representative evaluation and will become a testing ground for future methods of tabular DL.

## REPRODUCIBILITY STATEMENT

We describe our experimental setup in subsection 5.1 and Appendix C. The code is available at https://github.com/yandex-research/tabred

- Dataset downloading and preprocessing is handled by the provided code.
  Look for the `preprocessing` folder in the repository.
- Newly introduced datasets are avaialable at https://kaggle.com/TabReD
- Further instructions to reproduce experiments and plots are in the `README.md`

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

# A ADDITIONAL RESULTS AND ARTIFACTS

## A.1 EXPLORATION OF TABRED DATASETS: SHIFTS

In this section, we explore the temporal shift aspect of the TabReD datasets.

We plot standard deviations of MLP ensemble predictions over time together with model errors over the same timeframe. The plots are in Figure 2. These plots show a more nuanced view of the distribution shift in TabReD, and it's relationship to model performance:

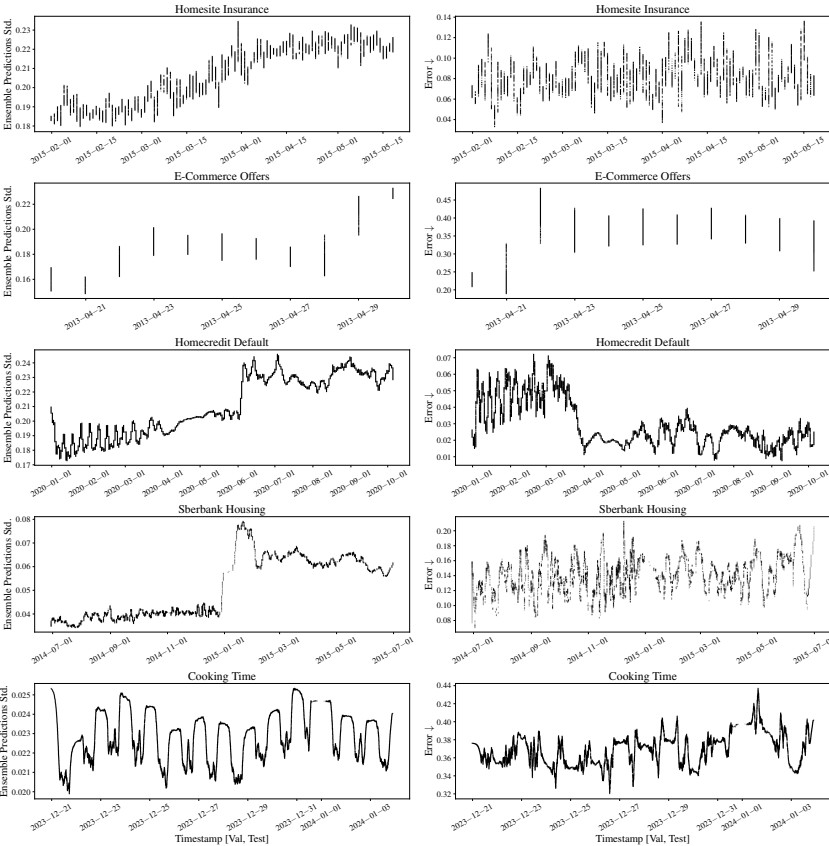

Figure 2: Relationship between distribution shift and performance on a subset of TabReD datasets. We use std in ensemble of MLP predictions as a proxy for the distribution shift. On the right side, we show errors (MAE for regression and error rate for binary classification)

- Some datasets exhibit trends where the shift increases over time and the error rate increases with it (this can be seen on Homesite Insurance and Sberbank Housing).

- In addition to global shift over time, some datasets exhibit strong seasonal behavior (e.g. there are specific times of day in the test set where performance drops significantly on the Cooking Time dataset).

- However, sometimes data shift might lead to seemingly better performance over time (e.g. as time goes by model predictions might improve). As variance and irreducible noise in the target variable can decrease over time, it could cause the performance metrics to improve over time. It is important to note that the model still suffers from the detrimental effects of the distributional shift, since if it was trained on the examples from the same domain as the one comprising the OOD test set, the performance would have been even better). The evolution of the degree of data shift and model errors on Homecredit Default is an example of such behavior.

Koh et al. (2021) define distribution shifts as a situation where "the training distribution differs from the test distribution". We would like to highlight that on our datasets, not only the distribution of some features changes from train to test, but these features are important for the models we used. We measure Wasserstein distance between train and test distributions of the target variable Y for the random and temporal data splits and an average Wasserstein distance between train and test for top-20 most important features. We can see that the distances between train and test are much more severe in the case of temporal split, which demonstrates by definition the existence of the distribution shift. We show our results in Figure 3.

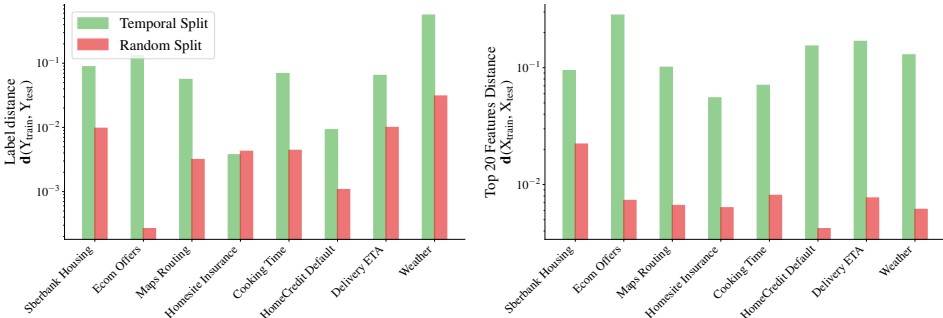

Figure 3: Wasserstein distance between train and test distributions of Y (left) and top-20 most important features (right). Green represents the case of temporal split, while red represents the random split. The distance is higher in the case of temporal split.

Per-dataset and model results comparing rankings and relative performance differences of methods on temporally and randomly split datasets are in Figure 4.

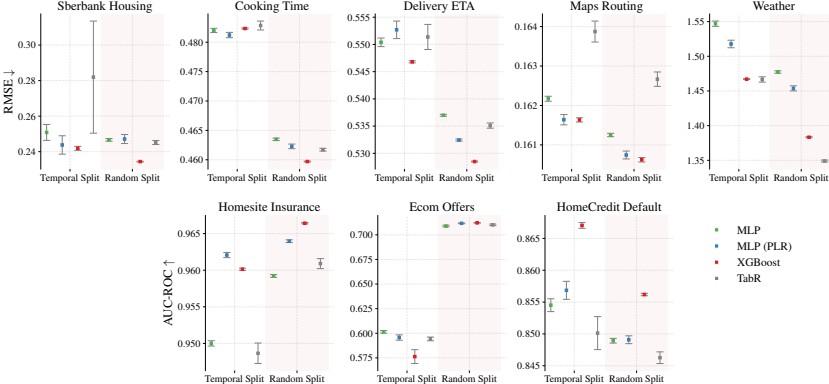

Figure 4: Comparison of performance on out-of-time and random in-domain test sets. The first row contains regression datasets, the metric is RMSE (lower is better). The second row contains binary classification datasets, the metric is AUC-ROC (higher is better). We can see the change in relative ranks and performance difference in addition to the overall performance drop. In particular, XGBoost lead decreases when comparing performance on task-appropriate time-shifted test sets.

**Temporal Shift's Influence on Performance**. We see that the spread of scores depending on random initialization for each model and data split is generally larger on the time-split based test set (most notable difference is on Sberbank Housing, Delivery ETA and Ecom Offers). This hints that temporal shift is present in the proposed datasets.

**Temporal Shift's Influence on Ranking and Relative Difference**. We can see that the ranking of different model categories and the spreads between model performance scores change when we use randomly split test sets. One notable example is XGBoost decreasing its performance margin to MLPs when evaluated on temporally shifted test sets (Sberbank Housing, Cooking Time, Delivery ETA, Weather, Ecom Offers and Quote Conversion datasets – most clearly seen comparing MLP-PLR with XGBoost). This might indicate that GBDTs are less robust to shifts, conversely performing

better on random splits, by possibly exploiting time-based leakage. Another notable example is TabR-S outperforming the baseline MLP (Cooking Time and Homesite Insurance) and even XGBoost (Weather).

## A.2 DISTRIBUTION SHIFT ROBUSTNESS METHODS

We evaluate two methods that aim to mitigate the effect of distribution shift. The first one is DeepCORAL (Sun & Saenko, 2016), we adapt the method to the temporal shift setting by bucketing timestamps into different domains, similar to Wild-Time (Yao et al., 2022). The second method is Deep Feature Reweighting (DFR) (Kirichenko et al., 2023), we adapt the method by finetuning the representation of the MLP baseline on the latter instances of the train dataset.

Table 5: Study of distribution shift robustness methods on TabReD.

| Methods | Classification (ROC AUC ↑) | | | Regression (RMSE ↓) | | | | | Average Rank |
|---|---|---|---|---|---|---|---|---|---|
| | Homesite Insurance | Ecom Offers | HomeCredit Default | Sberbank Housing | Cooking Time | Delivery ETA | Maps Routing | Weather | |
| MLP | 0.9500 | 0.6015 | 0.8545 | 0.2508 | 0.4820 | 0.5504 | 0.1622 | 1.5470 | 1.0 ± 0.0 |
| CORAL | 0.9498 | 0.6004 | 0.8549 | 0.2645 | 0.4821 | 0.5498 | 0.1622 | 1.5591 | 1.4 ± 0.7 |
| DFR | 0.9499 | 0.6013 | 0.8545 | 0.2494 | 0.4819 | 0.5515 | 0.1626 | 1.5513 | 1.4 ± 0.5 |

Both DFR and DeepCORAL do not improve upon the MLP baseline, in line with recent work by Gardner et al. (2023); Kolesnikov (2023) for other distribution shifts.

## A.3 EXPLORATION OF TABRED DATASETS: FEATURE CORRELATIONS

In this section, we explore the "feature-rich" aspect of TabReD datasets. For this, we plot the linear feature correlations and unary feature importances. To compute feature importances we use the mutual information in sklearn (Pedregosa et al., 2011) using methods from Kraskov et al. (2004).

Correlations together with feature-target mutual information for two benchmarks are in Figure 5.

Per-dataset and model results for the rich feature set vs small feature set comparison in subsection 5.4 is in Figure 6.

## B TABRED DATASET DETAILS

In this section, we provide more detailed dataset descriptions.

### B.1 DATASETS SOURCED FROM KAGGLE COMPETITIONS

Below, we provide short descriptions of datasets and corresponding tasks.

**Homesite Insurance**. This is a dataset from a Kaggle competition hosted by Homesite Insurance (Darrel, 2015). The task is predicting whether a customer will buy a home insurance policy based on user and insurance policy features (user, policy, sales and geographic information). Each row in the dataset corresponds to a potential [customer, policy] pair, the target indicates whether a customer bought the policy.

**Ecom Offers**. This is a dataset from a Kaggle competition hosted by the online book and game retailer DMDave (DMDave, 2014). The task in this dataset is a representative example of modeling customer loyalty in e-commerce. Concretely, the task is classifying whether a customer will redeem a discount offer based on features from two months' worth of transaction history. We base our feature engineering on one of the top solutions (MLWave, 2014).

**HomeCredit Default**. This is a second iteration of the popular HomeCredit tabular competition (Herman et al., 2024). The task is to predict whether bank clients will default on a loan, based on bank internal information and external information like credit bureau and tax registry data. This year competition focus was the model prediction stability over time. Compared to the more popular

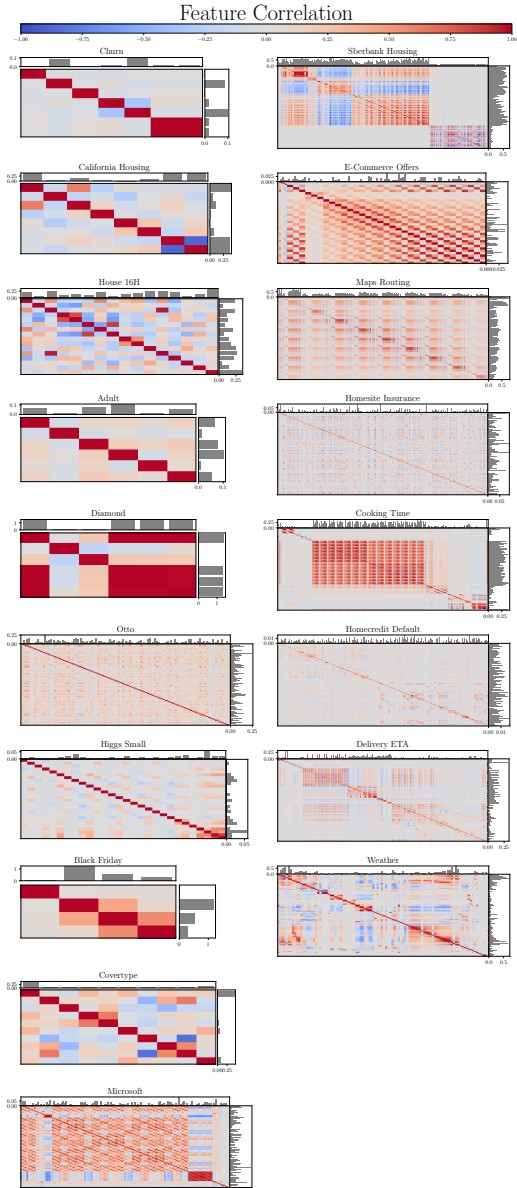

Figure 5: Feature correlations and importance via mutual information with target. On the left are datasets from Gorishniy et al. (2024), on the right are TabReD datasets. Datasets on the right are clearly more complex in terms of number of features and their correlation and importance patterns. The only comparably complex dataset on the left is Microsoft.

prior competition, this time there is more data and the timestamps are available. We base feature engineering and preprocessing code on top solutions (Kim, 2024).

**Sberbank Housing**. This dataset is from a Kaggle competition, hosted by Sberbank (Alexey Matveev, 2017). This dataset provides information about over 30000 transactions made in the Moscow housing market. The task is to predict the sale price of each property using the provided features describing each property condition, location, and neighborhood, as well as country economic indicators at the moment of the sale. We base our preprocessing code on discussions and solutions from the competition (Alijs & Johnpateha, 2017; Sidorova, 2017).

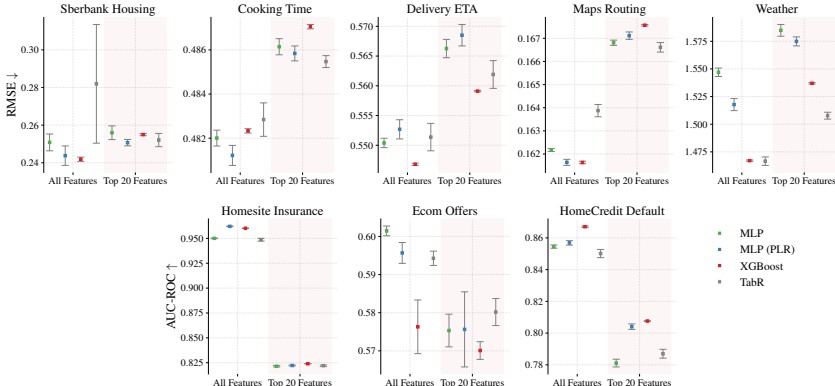

Figure 6: Comparison of performance on all features and top 20 features. The first row contains regression datasets, the metric is RMSE (lower is better). The second row contains binary classification datasets, the metric is AUC-ROC (higher is better). In additon to the expected overall performance drop we can see considerable improvement for the TabR model on datasets with less features, where it becomes competitive with XGBoost.

## B.2 NEW DATASETS FROM IN-THE-WILD ML APPLICATIONS

Here, we describe the datasets used by various ML applications that we publish with TabReD. All of these datasets were preprocessed for later use by a model in production ML systems. We apply deterministic transforms to anonymize the data for some datasets. We only publish the preprocessed data, as the feature engineering code and internal logs are proprietary. We will provide further details regarding licenses, preprocessing and data composition in the datasheet with supplementary materials upon acceptance (we can't disclose dataset details due to submission anonymity).

**Cooking Time**. For this dataset, the task is to determine how long it will take for a restaurant to prepare an order placed in a food delivery app. Features are constructed based on the information about the order contents and historical information about cooking time for the restaurant and brand, the target is a logarithm of minutes it took to cook the placed order.

**Delivery ETA**. For this dataset, the task is to determine the estimated time of arrival of an order from an online grocery store. Features are constructed based on the courier availability, navigation data and various aggregations of historical information for different time slices, the target is the logarithm of minutes it took to deliver an order.

**Maps Routing**. For this dataset, the task is to predict the travel time in the car navigation system based on the current road conditions. The features are aggregations of the road graph statistics for the particular route and various road details (like speed limits). The target is the logarithm of seconds per kilometer.

**Weather**. For this dataset, the task is weather temperature forecasting. This is a dataset similar to the one introduced in (Malinin et al., 2021), except it is larger, and the time-based split is available and used by default. The features are from weather station measurements and weather forecast physical models. The target is the true temperature for the moment in time.

## C    EXTENDED EXPERIMENTAL SETUP DESCRIPTION

This section is for extended information on the experimental setup.

## C.1    SOURCE CODE

We include source code for reproducing the results in the supplementary material archive with a brief `README.md` with instructions on reproducing the experiments. We also publish code for dataset preparation.

Table 6: Model ranks computed with Tamhane's T2 statistical test. Ranking in not significantly altered compared to our simple testing procedure.

| Methods | Classification (ROC AUC ↑) | | | Regression (RMSE ↓) | | | | |
|---|---|---|---|---|---|---|---|---|
| | Homesite Insurance | Ecom Offers | HomeCredit Default | Sberbank Housing | Cooking Time | Delivery ETA | Maps Routing | Weather |
| **Clasical ML Baselines** | | | | | | | | |
| XGBoost | 3 | 3 | **1** | **1** | 3 | 2 | **1** | 2 |
| LightGBM | 3 | 3 | 2 | 2 | 4 | 2 | 2 | **1** |
| CatBoost | 2 | 4 | 3 | 2 | 3 | **1** | 3 | 2 |
| RandomForest | 4 | 3 | 7 | 4 | 6 | 7 | 8 | 7 |
| Linear Model | 9 | 3 | 8 | 3 | 7 | 6 | 9 | 8 |
| **Deep Learning Methods** | | | | | | | | |
| MLP | 6 | **1** | 4 | 2 | 2 | 3 | 4 | 5 |
| SNN | 6 | **1** | 4 | 4 | 5 | 5 | 8 | 6 |
| DCNv2 | 8 | 2 | 6 | 4 | 5 | 4 | 9 | 7 |
| ResNet | 7 | **1** | 6 | 4 | 3 | 4 | 5 | 3 |
| FT-Transformer | **1** | 3 | 4 | **1** | 2 | 5 | 4 | 3 |
| MLP (PLR) | **1** | 2 | 4 | **1** | **1** | 4 | **1** | 4 |
| Trompt | 4 | 3 | 6 | 4 | 4 | 5 | 8 | 6 |
| MLP aug. | 5 | **1** | 6 | 4 | 4 | 5 | 6 | 4 |
| MLP aug. rec. | 5 | 2 | 9 | 3 | 5 | 5 | 7 | 4 |
| TabR | 6 | 2 | 5 | 4 | 4 | 3 | 7 | **1** |
| ModernNCA | 5 | 3 | 5 | 4 | 3 | 3 | 5 | 3 |
| CORAL | 6 | **1** | 4 | 4 | 2 | 3 | 4 | 6 |
| DFR | 6 | **1** | 4 | 2 | 2 | 4 | 5 | 5 |

## C.2 TUNING, EVALUATION, AND MODEL COMPARISONS

We replace NaN values with the mean value of a variable (zero after the quantile normalization). For categorical features with unmatched values in validation and test sets we encode such values as a special unknown category.

In tuning and evaluation setup, we closely follow the procedure described in (Gorishniy et al., 2021; 2024).

When comparing the models, we take standard deviations over 15 random initializations into account. We provide the ranks each method achieved below. In Table 3, following (Gorishniy et al., 2024), we rank method A below method B if $|B_{mean} - A_{mean}| < B_{stddev}$ and B score is better. To further demonstrate the statistical significance of our findings, we ran a Tamhane's T2 test of statistical significance for multiple comparisons. Results for Tamhane's test are in Table 6

We run hyperparameter optimization for 100 iterations for most models, the exceptions are FT-Transformer (which is significantly less efficient on datasets with hundreds of features) where we were able to run 25

For the exact hyperparameter search spaces, please see the source code. Tuning configs (`exp/**/tuning.toml`) together with the code are always the main sources of truth.

## C.3 ADDITIONAL IMPLEMENTATION DETAILS

We have taken each method's implementation from the respective official code sources, except for Trompt, which doesn't have an official code repository yet and instead was reproduced by us according to the information in the paper. We use default hyperparameters for the Trompt model from the paper (Chen et al., 2023b).

For the DFR (Kirichenko et al., 2023) baseline, we finetune the last layer on the last 20% of datapoints.

For CORAL (Sun & Saenko, 2016) we define domains by splitting instances based on a timestamp variable into 9 chunks.

## D   KAGGLE COMPETITIONS TABLE

We provide `annotated-kaggle-competitions.csv` with annotated Kaggle competitions in the supplementary repository. We used this table during sourcing the datasets from Kaggle. There are minimal annotations and notes. We annotate only tabular competitions with more than 500 competitors. We provide annotations for non-tabular datasets in this table.

## E   DISCUSSION OF INEFFECTIVENESS OF TIME-SERIES AND FORECASTING METHODS ON TABRED

Discussion of timeshift-agnostic vs forecasting methods

We believe there is a separation between forecasting tasks and our non-I.I.D. tabular prediction tasks. For our datasets, we specifically chose datasets with rich features, where the time feature does not overwhelm the importance of other features. In this regard, our task is different from time-series tasks. When looking at the discussion section on Kaggle, we notice that none of the competitions from which we source our data have a competitive solution based on forecasting. Here are the details:

- On Sberbank Housing dataset, the only mention of forecasting is for feature engineering (Kaggle forum discussion). Even in that context, forecasting was not successful and failed to improve performance of the downstream model

- On Homesite Insurance dataset, the only mention of utilizing time-series is again for feature engineering, and again it only gives an insignificant boost to the score of 2e-4 (Kaggle forum discussion). All top decisions mentioned in discussion utilize shift-agnostic models such as XGBoost.

- On the E-Commerse dataset, time and forecasting is not mentioned at all, and the top solution focuses on feature engineering approaches (Kaggle forum discussion).

- On HomeCredit dataset, the mistake in metric choice by hosts resulted in some conditions using unrealistic hacks to win. Nevertheless, in the discussion process, competitors agreed, than when evaluation by pure AUC, feature engineering is more critical (Kaggle forum discussion).

- For our newly introduced datasets, no forecasting experiments have proved useful in production, and simple shift-agnostic tabular ML models have proven to be useful in real-world applications scenarios.

## F   DETAILED ACADEMIC DATASETS OVERVIEW

In this section, we go through each dataset, and list its problems and prior uses in literature. We specify whether time-based splits should preferably be used for the dataset and whether it is available (e.g. datasets come with timestamps).

We also adhere to the definition of tabular dataset proposed by Kohli et al. (2024) and annotate the existing datasets with one of five categories: raw data (the least tabular type), homogeneously extracted features HomE (e.g. similar features from one source, like image descriptors or features), heterogeneous extracted features HetE (different concepts, still extracted from one source) and semi-tabular/tabular (when there are multiple sources for HetE features).

The full table with annotation is available in the root of the code repository in the `academic-datasets-summary.csv` file.

We also provide commentary and annotations directly in the appendix below.

## 100-PLANTS-TEXTURE

**Tags**: HomE

**#Samples**: 1599     **#Features**: 65     **Year**: 2012

**Comments**: This is a small dataset with images and image-based features. The dataset first appeared in the paper "Plant Leaf Classification Using Probabilistic Integration of Shape, Texture and Margin Features. Signal Processing, Pattern Recognition and Applications", written by Mallah et al in 2013, and contains texture features extracted from the images of the leaves taken from Royal Botanic Gardens in Kew, UK. The task at hand is to recognize which leave is being described by the given features. While this feature extraction was a beneficial way to handle vision-based information in the early 2010s, modern approaches to CV focus on the specific architectures better suited for image data.

## ALOI

**Tags**: HomE

**#Samples**: 108000     **#Features**: 128     **Year**: 2005

**Comments**: The dataset describes a collection of images provided by Geusebroek et al. The features used in this version are color histogram values.

## ADULT

**Tags**: Tabular, Timesplit Needed

**#Samples**: 48842     **#Features**: 33     **Year**: 1994

**Comments**: One of the most popular tabular datasets, Adult was created by Barry Becker based on the 1994 Census database. The target variable is a binary indicator of whether a person has a yearly income above 50000$.

## AILERONS

**Tags**: Raw

**#Samples**: 13750     **#Features**: 40     **Year**: 2014

**Comments**: This data set addresses a control problem, namely flying an F16 aircraft. The attributes describe the status of the aeroplane, while the goal is to predict the control action on the ailerons of the aircraft. According to the descriptions available, the dataset first appears in a collection of regression datasets by Luis Torgo and Rui Camacho made in 2014, but the original website with the description (ncc.up.pt/ ltorgo/Regression/DataSets.html) seems to no longer respond. The task of controlling a vehicle through machine learning has gained a large interest in recent years, but it is not done through tabular machine learning, instead often utilizing RL and using wider range of sensors than those used in the non-self-driving version of the vehicle.

## AUSTRALIAN

**Tags**: Tabular, Timesplit Needed

**#Samples**: 690     **#Features**: 15     **Year**: 1987

**Comments**: Anonymized credit approval dataset. Corresponds to a real-life task, but is very small, with only 15 features, and no way to create a time/or even user-based validation split, no time variable is available

## BIKE_SHARING_DEMAND

**Tags**: Leak, Tabular, Timesplit Needed, Timesplit Possible

**#Samples**: 17379     **#Features**: 6     **Year**: 2012

**Comments**: This dataset was produced based on the data from the Capital Bikeshare system from 2011 and 2012. The task is to predict the count of bikes in use based on time and weather conditions. No forecasting FE is done, only weather and date are available, forecasting tasks in the real-world, if solved by tabular models involve extensive feature engineering. While the task is to predict demand at a specific time, the time-based split is not performed, although it is possible. Due to the random i.i.d. split of the dataset, while predicting on a test object models could use information from the train examples close in time to the test one, which wouldn't be possible in real-life conditions, since a model is used after it is trained.

## BIORESPONSE

**Tags**: HomE

**#Samples**: 3751     **#Features**: 1777     **Year**: 2012

**Comments**: These datasets present a classification problem on molecules. The underlying data is not tabular, graph-based methods, incorporating 3d structure are known to outperform manual descriptors (https://ogb.stanford.edu/docs/lsc/leaderboards/#pcqm4mv2), this is not a mainstream task in its formulation. For more up to date, datasets and classification tasks on molecules one could use https://moleculenet.org for example

## BLACK FRIDAY

**Tags**: Timesplit Needed

**#Samples**: 166821     **#Features**: 9     **Year**: 2019

**Comments**: No time split, predicting customer's purchase amount from demographic features. "A retail company "ABC Private Limited" wants to understand the customer purchase behaviour (specifically, purchase amount) against various products of different categories. They have shared purchase summaries of various customers for selected high-volume products from last month. The data set also contains customer demographics (age, gender, marital status, city_type, stay_in_current_-city), product details (product_id and product category) and Total purchase_amount from last month." No way to check if the dataset is real. Potential leakage. The task looks artificial we need to predict the purchase amount based on 6 users and 3 product features (there are 5k users and 3k unique users and products on 160k samples, the mean price for a product is a strong baseline).

## BRAZILIAN_HOUSES

**Tags**: Tabular, Timesplit Needed

**#Samples**: 10692     **#Features**: 8     **Year**: 2020

**Comments**: Similar to other housing market prediction tasks, the data is a snapshot of listings on a Brazilian website with houses to rent. No way to create a time split.

## BROKEN MACHINE

**Tags**: Synthetic or Untraceable, Tabular

**#Samples**: 900000     **#Features**: 58     **Year**: 2021?

**Comments**: This dataset was not described anywhere, and the link to the original publication on Kaggle is no longer working.

## CALIFORNIA HOUSING

**Tags**: Tabular, Timesplit Needed

**#Samples**: 20640     **#Features**: 8     **Year**: 1990

**Comments**: This data comes from 1990 US Census data. Each object describes a block group, which on average includes 1425.5 individuals. The features include information about housing units inside a block group, as well as median income reported in the area. The target variable is the ln of a median house price. Due to the fact that the target variable is averaged across a large number of houses, KNN algorithms using the coordinates of a block are very effective. Housing prices are quickly changing in time, which presents an additional challenge for tabular ML models, however, on this dataset a time-based split is impossible. The provided features are also shallow in comparison to a dataset that may be used in an industrial scenario, e.g. housing dataset included in our publication includes hundreds of features as opposed to this dataset, which includes only 8.

## CHURN MODELLING

**Tags**: Synthetic or Untraceable, Tabular, Timesplit Needed

**#Samples**: 10000     **#Features**: 11     **Year**: 2020

**Comments**: This dataset describes a set of customers of a bank, with a task of classifying whether a user will stay with the bank. Not a time split. Unknown source (may be synthetic). Not rich information. Narrow, No License. No canonical split (No time dimension)

## EPSILON

**Tags**: Synthetic or Untraceable, Tabular

**#Samples**: 500000     **#Features**: 2000     **Year**: 2008

**Comments**: This dataset comes from a 2008 competition "Large Scale Learning Challenge" by the K4all foundation. The source of the data is unclear, the dataset might be synthetic.

## FACEBOOK COMMENTS VOLUME

**Tags**: Leak, Tabular, Timesplit Needed

**#Samples**: 197080     **#Features**: 51     **Year**: 2016

**Comments**: This dataset presents information about a facebook post and the target is to determine how many comments will appear within a period of time. Leakage. Same comments from different points in time, random split is inappropriate. This case is described in the appendix of a TabR paper, as this model was able to exploit the leak do get extreme performance improvements

## GESTURE PHASE

**Tags**: Leak, Tabular, Timesplit Needed, Timesplit Possible

**#Samples**: 9873     **#Features**: 32     **Year**: 2016

**Comments**: The task of this dataset is to classify gesture phases. Features are the speed and the acceleration from kinect. There are 7 videos from 3 users (3 gesture sequences from 2 and one from an additional user). The paper, which introduced the dataset mentions that using the same user (but a different story) for evaluation influences the score. Tabular DL papers, use random split on this dataset – this is not assessing the performance on new users, not even on new sequences of one user, not a canonical split. Without canonical split, the task contains leakage, which is easily exploited by using retrieval methods or overtuning models.

## HELENA

**Tags**: Synthetic or Untraceable, HetE

**#Samples**: 65196     **#Features**: 27     **Year**: 2018

**Comments**: The data was provided by AutoML challenge, and the dataset was created from objects from another domain, such as text, audio, or video, compressed into tabular form.

### HIGGS

**Tags**: Tabular

**#Samples**: 940160   **#Features**: 24   **Year**: 2014

**Comments**: Physics simulation data.

### HOUSE 16H

**Tags**: Tabular, Timesplit Needed

**#Samples**: 22784   **#Features**: 16   **Year**: 1990

**Comments**: No time features, comes from the US Census 1990. Feature selection was performed, non correlated features were selected for house 16H(hard). Narrow, by definition, less important features. Learning problem, not the most representative for the real world task

### JANNIS

**Tags**: HetE

**#Samples**: 83733   **#Features**: 54   **Year**: 2018

**Comments**: The data was provided by AutoML challenge, and the dataset was created from objects from another domain, such as text, audio, or video, compressed into tabular form.

### KDDCUP09_UPSELLING

**Tags**: Tabular, Timesplit Needed

**#Samples**: 5032   **#Features**: 45   **Year**: 2009

**Comments**: Real-world data and problem from Orange telecom company, the taks is binary classification of upselling. All variables are anonyimzed, the time is not available (but predictions in this problem do happen in the future) only i.i.d train with labels is available

### MAGICTELESCOPE

**Tags**: Tabular

**#Samples**: 13376   **#Features**: 10   **Year**: 2004

**Comments**: Physics simulation

### MERCEDES_BENZ_GREENER_MANUFACTURING

**Tags**: Tabular

**#Samples**: 4209   **#Features**: 359   **Year**: 2017

**Comments**: This dataset presents features about a Mercedes car, the task is to determine the time it will take to pass testing.

### MIAMIHOUSING2016

**Tags**: Tabular, Timesplit Needed, Timesplit Possible

**#Samples**: 13932   **#Features**: 14   **Year**: 2016

**Comments**: The dataset comes from publicly available information on house sales in Miami in 2016. While this dataset has several improvements when compared to the california housing dataset, such as not averaging prices in a block, as well as availability of date of sale, the features are still shallow when compared to the housing dataset presented in this paper.

### MINIBOONE

**Tags**: Tabular

**#Samples**: 72998    **#Features**: 50    **Year**: 2005

**Comments**: Physics simulation

### ONLINENEWSPOPULARITY

**Tags**: Timesplit Needed

**#Samples**: 39644    **#Features**: 59    **Year**: 2015

**Comments**: This dataset contains information about articles published by Mashable, and posits a task of predicting number of shared of each article. Features are mostly NLP related, e.g. LDA and number of specific keywords. Would be better solved by NLP approaches.

### OTTO GROUP PRODUCTS

**Tags**: Tabular, Timesplit Needed, Timesplit Possible

**#Samples**: 61878    **#Features**: 93    **Year**: 2015

**Comments**: This data comes from 2015 kaggle competition hosted by The Otto Group. The objective is to classify a product's category. Each row corresponds to a single product. There are a total of 93 numerical features, which represent counts of different events. All features have been obfuscated and will not be defined any further. No time meta-feature, no way to ensure there is no time leak (what are the events? what if the distribution of these counts shifts over time?). No canonical split available, no details on the competition website on the nature of the features

### SGEMM_GPU_kernel_performance

**Tags**: Leak, Tabular

**#Samples**: 241600    **#Features**: 9    **Year**: 2018

**Comments**: Leakage. The task is to predict the time that it takes to multiply two matrices, but 3 out of 4 target variables are given. With them included, all other features have zero random forest importance.

### SANTANDER CUSTOMER TRANSACTIONS

**Tags**: Tabular, Timesplit Needed

**#Samples**: 200000    **#Features**: 200    **Year**: 2019

**Comments**: The data comes from 2019 kaggle competition by Santander. The task is to predict whether a customer will make a specific transaction. Performed processing is unknown. Time-based split is appropriate but not possible to perform.

### SHIFTS WEATHER (IN-DOMAIN-SUBSET)

**Tags**: Leak, Tabular, Timesplit Possible

**#Samples**: 397099    **#Features**: 123    **Year**: 2021

**Comments**: The dataset first appeared in the 2021 paper concerning distributional shift. Leakage. In-domain version used. Samples from the future used for prediction. Retrieval methods such as TabR achieve large performance improvements.

### SPEEDDATING

**Tags**: Tabular, Timesplit Needed

**#Samples**: 8378     **#Features**: 121     **Year**: 2004

**Comments**: This dataset describes experimental speed dating events that took place from 2002 to 2004. The data describes the responses of participants to a questionnaire, and the target variable is whether they matched or not.

### TABLESHIFT ASSISTMENTS

**Tags**: Tabular, Timesplit Needed

**#Samples**: 2600000     **#Features**: 16     **Year**: 2013

**Comments**: Predict whether the student answers correctly. Features include: student-, problem-, and school-level features, the dataset also contains affect predictions for students based on an experimental affect detector implemented in ASSISTments. Timesplit is not possible.

### TABLESHIFT CHILDHOOD LEAD

**Tags**: Tabular, Timesplit Needed, Timesplit Possible

**#Samples**: 27000     **#Features**: 8     **Year**: 2023

**Comments**: The data comes from CDC National Health and Nutrition Examination Survey, and the task in this dataset is to predict whether a person has high blood lead levels based on answers to a questionnaire.

### TABLESHIFT COLEGE SCORECARD

**Tags**: Tabular, Timesplit Needed

**#Samples**: 124699     **#Features**: 119     **Year**: 2023

**Comments**: The task is to predict the completion rate for a college. The College Scorecard is an institution-level dataset compiled by the U.S. Department of Education from 1996-present

### TABLESHIFT DIABETES

**Tags**: Tabular, Timesplit Needed, Timesplit Possible

**#Samples**: 1444176     **#Features**: 26     **Year**: 2021

**Comments**: Determine Diabetes diagnosis from a telephone survey. We use data provided by the Behavioral Risk Factors Surveillance System (BRFSS). BRFSS is a large-scale telephone survey conducted by the Centers of Disease Control and Prevention.

### TABLESHIFT FOOD STAMPS

**Tags**: Tabular, Timesplit Needed, Timesplit Possible

**#Samples**: 840582     **#Features**: 21     **Year**: 2023

**Comments**: Data source bias (US based surveys), comes from ACS. Narrow. No time split provided in the benchmark version.

### TABLESHIFT HELOC

**Tags**: Tabular, Timesplit Needed

**#Samples**: 10000     **#Features**: 23     **Year**: 2018

**Comments**: TableShift uses the Home Equity Line of Credit (HELOC) Dataset from the FICO Explainable Machine Learning Challenge

### TABLESHIFT HYPERTENTION

**Tags**: Tabular, Timesplit Needed, Timesplit Possible

**#Samples**: 846000    **#Features**: 14    **Year**: 2021

**Comments**: Determine whether a person has hypertension from a telephone survey. We use data provided by the Behavioral Risk Factors Surveillance System (BRFSS). BRFSS is a large-scale telephone survey conducted by the Centers of Disease Control and Prevention.

### TABLESHIFT ICU HOSPITAL MORTALITY

**Tags**: Semi- Tabular, Timesplit Needed

**#Samples**: 23944    **#Features**: 7520    **Year**: 2016

**Comments**: The data comes from MIMIC-III, describing records from Beth Israel Deaconess Medical Center. The data used in this dataset would be more effectively processed as time series and sequences.

### TABLESHIFT ICU LENGTH OF STAY

**Tags**: Semi- Tabular, Timesplit Needed

**#Samples**: 23944    **#Features**: 7520    **Year**: 2016

**Comments**: The data comes from MIMIC-III, describing records from Beth Israel Deaconess Medical Center. The data used in this dataset would be more effectively processed as time series and sequences.

### TABLESHIFT INCOME

**Tags**: Tabular, Timesplit Needed, Timesplit Possible

**#Samples**: 1600000    **#Features**: 15    **Year**: 2018

**Comments**: The task is to predict person's income based on their answers to a survey. Data is provided by American Community Survey.

### TABLESHIFT PUBLIC COVERAGE

**Tags**: Tabular, Timesplit Needed, Timesplit Possible

**#Samples**: 5900000    **#Features**: 11    **Year**: 2018

**Comments**: The task is to predict whether a person is covered by public health insurance based on their answers to a survey. Data is provided by American Community Survey.

### TABLESHIFT READMISSION

**Tags**: Tabular, Timesplit Needed

**#Samples**: 99000    **#Features**: 47    **Year**: 2008

**Comments**: "This study used the Health Facts database (Cerner Corporation, Kansas City, MO), a national data warehouse that collects comprehensive clinical records across hospitals throughout the United States." Clinical patient with diabetes data. 47 features with questionnaire like information (num_previous visits, which medication patients were using, which diagnosis patients had). No time feature available.

### TABLESHIFT SEPSIS

**Tags**: Semi- Tabular, Timesplit Needed

**#Samples**: 1500000    **#Features**: 41    **Year**: 2019

**Comments**: Predict whether a person will develop sepsis in the next 6 months based on the data about their health, including questionnaire answers and patient records.

### TABLESHIFT UNEMPLOYMENT

**Tags**: Tabular, Timesplit Needed, Timesplit Possible

**#Samples**: 1700000    **#Features**: 18    **Year**: 2018

**Comments**: The task is to predict whether a person is unemployed based on their answers to a survey. Data is provided by American Community Survey.

### TABLESHIFT VOTING

**Tags**: Tabular, Timesplit Needed, Timesplit Possible

**#Samples**: 8000    **#Features**: 55    **Year**: 2020

**Comments**: The prediction target for this dataset is to determine whether an individual will vote in the U.S presidential election, from a detailed questionnaire. It seems like the data goes all the way back to 1948, which makes this not realistic when not using time split

### VESSEL POWER R

**Tags**: Tabular, Timesplit Needed, Timesplit Possible

**#Samples**: 554642    **#Features**: 10    **Year**: 2022

**Comments**: The dataset describes information about a shipping line, with the task of determining how much power is needed.

### VESSEL POWER S

**Tags**: Synthetic or Untraceable, Tabular, Timesplit Needed, Timesplit Possible

**#Samples**: 546543    **#Features**: 10    **Year**: 2022

**Comments**: Synthetic version of Vessel Power dataset

### YEAR

**Tags**: HomE

**#Samples**: 515345    **#Features**: 90    **Year**: 2011

**Comments**: This dataset describes musical compositions, with the target variable being a year in which the composition was created. Another domain (audio features - extracted from audio, thus suitable for tabular DL, but DL for audio on raw data is preferable in this domain. Year prediction, solved as a regression task. Dataset does not correspond to a real-world problem (the year meta-data is easy to obtain, no need for prediciton). Problem with the formulation – solving as a classification problem might be preferable (98

### ADA-AGNOSTIC

**Tags**: Tabular, Timesplit Needed

**#Samples**: 4562    **#Features**: 49    **Year**: 1994

**Comments**: This dataset is a processed version of the popular Adult dataset. This particular rendition of the well known dataset first appeared in the competition "Agnostic Learning vs. Prior Knowledge" that took place at IJCNN 2007. The differences with the original Adult include some features or categorical values being dropped and missing values being preprocessed. Overall, this rendition is

plagued by the same problems that the original Adult dataset has, making it serve as a duplicate less useful for analysing tabular machine learning in the context of large benchmarks.

### AIRLINES

**Tags**: Tabular, Timesplit Needed

**#Samples**: 539382     **#Features**: 8     **Year**: 2006

**Comments**: The airlines dataset was created for the Data Expo competition in 2006 by Elena Ikonomovska. Unfortunately, the competition link provided in the secondary sources does not work anymore. The proposed task for the dataset is to predict flight delays based on Airline, flight number, time, source and destination. While the data is sourced in the real world, and the task of predicting the delay of the flight certainly could be solved with tabular deep learning, the provided features lack most information essential to predicting the delay. Time based train/val/test split is important but impossible to produce with this dataset.

### ALBERT

**Tags**: Synthetic or Untraceable, HetE

**#Samples**: 425240     **#Features**: 79     **Year**: 2018

**Comments**: This is an anonymized dataset with unknown origin. Based on the source description, the original data could be of any modality. There is no way to control a train / test split without task details.

### ANALCATDATA_SUPREME

**Tags**: Tabular, Timesplit Needed, Timesplit Possible

**#Samples**: 4052     **#Features**: 7     **Year**: 2003

**Comments**: The analcatdata_supreme dataset first appeared in the 2003 book "Analysing Categorical Data" by Jeffrey S. Simonoff. This dataset contains a collection of decisions made by the Supreme Court of the United Stated from 1953 to 1988. The information used is very shallow, and the data was introduced for domain specific analysis, not to compare performance on random splits of the dataset

### ARTIFICIAL-CHARACTERS

**Tags**: Leak, Synthetic or Untraceable, Raw

**#Samples**: 10218     **#Features**: 8     **Year**: 1993

**Comments**: This database has been artificially generated. It describes the structure of the capital letters A, C, D, E, F, G, H, L, P, R, indicated by a number 1-10, in that order (A=1,C=2,...). Each letter's structure is described by a set of segments (lines) which resemble the way an automatic program would segment an image. The dataset consists of 600 such descriptions per letter.

Originally, each 'instance' (letter) was stored in a separate file, each consisting of between 1 and 7 segments, numbered 0,1,2,3,... Here they are merged. That means that the first 5 instances describe the first 5 segments of the first segmentation of the first letter (A). Also, the training set (100 examples) and test set (the rest) are merged. The next 7 instances describe another segmentation (also of the letter A) and so on.

Not a tabular data task (synthetic letter classification). When used as a tabular dataset, leak could easily be exploited through the "V7: diagonal, this is the length of the diagonal of the smallest rectangle which includes the picture of the character. The value of this attribute is the same in each object."

### AUDIOLOGY

**Tags**: Tabular, Timesplit Needed

**#Samples**: 226     **#Features**: 70     **Year**: 1987

**Comments**: The audiology dataset has been provided by Professor Jergen at Baylor College of Medicine in 1987, and contains information describing the hearing ability of different patients

### BALANCE-SCALE

**Tags**: Synthetic or Untraceable

**#Samples**: 625     **#Features**: 5     **Year**: 1994

**Comments**: This data set was generated to model psychological experimental results. Each example is classified as having the balance scale tip to the right, tip to the left, or be balanced. The attributes are the left weight, the left distance, the right weight, and the right distance. The correct way to find the class is the greater of (left-distance * left-weight) and (right-distance * right-weight). If they are equal, it is balanced. This is not a real world problem. Just the data from psychological study, easily solvable with one equation

### BANK-MARKETING

**Tags**: Tabular, Timesplit Needed

**#Samples**: 10578     **#Features**: 7     **Year**: 2010

**Comments**: Dataset describes 17 marketing campaigns by a bank from 2008 to 2010. A set of features is not very rich, but reasonable (ideally there would be more user features and statistics).

### CNAE-9

**Tags**: HomE

**#Samples**: 1080     **#Features**: 857     **Year**: 2009

**Comments**: This dataset only offers the frequencies of 800 words as features, the data is purely from the NLP domain

### COLIC

**Tags**: Tabular, Timesplit Needed

**#Samples**: 368     **#Features**: 27     **Year**: 1989

**Comments**: The dataset of horses symptoms and whether or not they required surgery.

### COMPASS

**Tags**: Leak, Tabular, Timesplit Needed

**#Samples**: 16644     **#Features**: 17     **Year**: 2017

**Comments**: This dataset's task is to determine whether a person will be arrested again after their release based on simple statistical features. The dataset first appeared in the paper "It's COMPASlicated: The Messy Relationship between RAI Datasets and Algorithmic Fairness Benchmarks" by Bao et al. Seemingly there are a lot of duplicates in the data, which leads to leakage when the random split is applied. Retrieval methods such as TabR achieve large performance gains.

### COVERTYPE

**Tags**: Tabular, Timesplit Needed

**#Samples**: 423680     **#Features**: 54     **Year**: 1998

**Comments**: This dataset comes from 1998 study comparing different methods for predicting forest cover types from cartographic variables. No time features are included. Not representative of a

real-world task: predicting forest cover-type solely from geological and cartographic features comes up less frequently, than directly processing GNSS data

## CPU_ACT

**Tags**: Tabular, Timesplit Needed

**#Samples**: 8192    **#Features**: 21    **Year**: 1999

**Comments**: This data represents logs from a server computer. The task is to predict the portion of time that cpu runs in user mode.

## CREDIT

**Tags**: Tabular, Timesplit Needed

**#Samples**: 16714    **#Features**: 10    **Year**: 2011

**Comments**: Dataset from the kaggle competition hosted by "Credit Fusion". Corresponds to a real-world prediction problem. Not possible to create an out-of-time evaluation set. Relatively (relative to the modern dataset, e.g. https://www.kaggle.com/competitions/amex-default-prediction/overview) few features available.

## CREDIT-APPROVAL

**Tags**: Tabular, Timesplit Needed

**#Samples**: 690    **#Features**: 16    **Year**: 1987

**Comments**: Same as Australian (but without preprocessing)

## CREDIT-G

**Tags**: Tabular, Timesplit Needed

**#Samples**: 10000    **#Features**: 21    **Year**: 1994

**Comments**: This dataset includes a number of simple features useful for determining whether the bank can expect a return on a credit. The nature of labels is not explained, time feature is not used

## DIAMONDS

**Tags**: Tabular, Timesplit Needed

**#Samples**: 53940    **#Features**: 9    **Year**: 2015

**Comments**: The exact source of the data is unclear. The task is to predict the price of a diamond by its characteristics. Diamond prices fluctuate in time, however no timestamp information is available.

## ELECTRICITY

**Tags**: Leak, Tabular, Timesplit Needed, Timesplit Possible

**#Samples**: 38474    **#Features**: 7    **Year**: 1998

**Comments**: Data comes from the Australian New South Wales Electricity Market. The task is to predict whether electricity prices will go up or down. When a random split is used, there is a leak in the data, and retrieval methods such as TabR can achieve near 100% accuracy.

## ELEVATORS

**Tags**: Raw

**#Samples**: 16599    **#Features**: 16    **Year**: 2014

**Comments**: This data set addresses a control problem, namely flying an F16 aircraft. The attributes describe the status of the aeroplane, while the goal is to predict the control action on the ailerons of the aircraft. According to the descriptions available, the dataset first appears in a collection of regression datasets by Luis Torgo and Rui Camacho made in 2014, but the original website with the description (ncc.up.pt/ ltorgo/Regression/DataSets.html) seems to no longer respond. The task of controlling a vehicle through machine learning has gained a large interest in recent years, but it is not done through tabular machine learning, instead often utilizing RL and using a wider range of sensors than those used in the non-self-driving version of the vehicle.

### EYE_MOVEMENTS

**Tags**: Leak, Tabular

**#Samples**: 7608    **#Features**: 20    **Year**: 2005

**Comments**: Time-series, Grouped data. This is a grouped dataset, some models are able to find a leak and predict based on an assignment number perfectly (MLP-PLR for example).

### FIFA

**Tags**: Tabular, Timesplit Needed

**#Samples**: 18063    **#Features**: 5    **Year**: 2021

**Comments**: This dataset contains information about FIFA soccer players in 2021, and the target variable is their wages. The provided features include age, weight, height, and information about time spent in the player's club, as well as the price in release clause. This dataset does not correspond to any real-world task, and the provided features are very shallow, as they luck any information about a player's performance in previous games

### GUILLERMO

**Tags**: HetE

**#Samples**: 20000    **#Features**: 4297    **Year**: 2018

**Comments**: The data was provided by AutoML challenge, and the dataset was created from objects from another domain, such as text, audio, or video, compressed into tabular form.

### HEART-H

**Tags**: Tabular, Timesplit Needed

**#Samples**: 294    **#Features**: 14    **Year**: 1988

**Comments**: This dataset was originally created by Andras Janosi et al. in 1988. A very small dataset including features describing person's questionnaire responses as well as some compressed test results. Statistics are too shallow to adequately solve the task at hand.

### HOUSE_SALES

**Tags**: Tabular, Timesplit Needed, Timesplit Possible

**#Samples**: 21613    **#Features**: 15    **Year**: 2015

**Comments**: This dataset was created based on public records of house sales from May 2014 to May 2015. While this dataset has several improvements when compared to the California housing dataset, such as not averaging prices in a block, as well as availability of date of sale, the features are still shallow when compared to the housing dataset presented in this paper.

### HOUSES

**Tags**: Tabular, Timesplit Needed

**#Samples**: 20640    **#Features**: 8    **Year**: 1990

**Comments**: Data source bias, repeated dataset (unknown source in the description, but this is literally california_housing with a different name and two features slightly altered)

### ISOLET

**Tags**: HomE

**#Samples**: 7797    **#Features**: 613    **Year**: 1994

**Comments**: The dataset describes features extracted from audio recordings of the name of each letter of the English alphabet. The task is to classify the phoneme. This task would be better solved by raw audio processing.

### JASMINE

**Tags**: Synthetic or Untraceable, HetE

**#Samples**: 2984    **#Features**: 145    **Year**: 2018

**Comments**: The data was provided by AutoML challenge, and the dataset was created from objects from another domain, such as text, audio, or video, compressed into tabular form.

### JUNGLE-CHESS

**Tags**: Synthetic or Untraceable, Raw

**#Samples**: 44819    **#Features**: 7    **Year**: 2014

**Comments**: Game simulation, not a real ML task.

### KC1

**Tags**: HetE

**#Samples**: 2109    **#Features**: 22    **Year**: 2004

**Comments**: This dataset was created by Mike Chapman at NASA, and it contains features associated with the software quality. The task is to predict whether the code has any defects. Nowadays, the task of code quality analysis is solved mainly using NLP methods and is not tabular.

### KDD_IPUMS_LA_97-SMALL

**Tags**: Tabular, Timesplit Needed

**#Samples**: 5188    **#Features**: 20    **Year**: 1997

**Comments**: The data is a subsample of census responses from the Los Angeles area for years 1970, 1980 and 1990. Unknown target variable (some categorical column from census binarized). Not a real-world task, based on census data.

### LYMPH

**Tags**: Tabular, Timesplit Needed

**#Samples**: 148    **#Features**: 19    **Year**: 1988

**Comments**: This dataset was collected in November 1988 for University Medical Centre, Institute of Oncology, Ljubljana, Yugoslavia by Bojan Cestnik. It includes results of lymph test. The task is to classify lymph in one of four categories. Unfortunately, the dataset contains only 2 samples with normal lymph, making it hard for the dataset to be used for training a real-world model categorizing lymph.

### MEDICAL_CHARGES

**Tags**: Tabular, Timesplit Needed

**#Samples**: 163065    **#Features**: 3    **Year**: 2019

**Comments**: Public medicare data from 2019. According to openml analysis, only one of the features is important for prediction.

### MFEAT-FOURIER

**Tags**: HomE

**#Samples**: 2000    **#Features**: 77    **Year**: 1998

**Comments**: One of a set of 6 datasets describing features of handwritten numerals (0 - 9) extracted from a collection of Dutch utility maps.

### MFEAT-ZERNIKE

**Tags**: HomE

**#Samples**: 2000    **#Features**: 48    **Year**: 1998

**Comments**: One of a set of 6 datasets describing features of handwritten numerals (0 - 9) extracted from a collection of Dutch utility maps.

### MONKS-PROBLEMS-2

**Tags**: Synthetic or Untraceable

**#Samples**: 601    **#Features**: 7    **Year**: 1992

**Comments**: Simple toy synthetic, the task of determining whether there are exactly two ones among the 6 binary variables.

### NOMAO

**Tags**: Tabular

**#Samples**: 34465    **#Features**: 119    **Year**: 2013

**Comments**: Active learning dataset, the task is determining whether two geo-location points are the same. Hand-labeled by an expert of Nomao.

### NYC-TAXI-GREEN-DEC-2016

**Tags**: Tabular, Timesplit Needed

**#Samples**: 581835    **#Features**: 9    **Year**: 2016

**Comments**: The data was provided by the New York City Taxi and Limousine Commission, and the task is to predict tip amount based on simple features describing the trip.

### PARTICULATE-MATTER-UKAIR-2017

**Tags**: Tabular, Timesplit Needed, Timesplit Possible

**#Samples**: 394299    **#Features**: 6    **Year**: 2017

**Comments**: Hourly particulate matter air pollution data of Great Britain for the year 2017. Time features available, prior work uses random split. There are only 6 features, describing time and location. This is a time-series forecasting problem (2 features from the original dataset missing). This is more likely a time-series problem, as there are not many heterogeneous features related to the task, only time-based features

### PHONEME

**Tags**: HomE

**#Samples**: 5404    **#Features**: 6    **Year**: 1993

**Comments**: The dataset describes a collection of phonemes and presents a task of classifying between nasal and oral sounds. The phonemes are transcribed as follows: sh as in she, dcl as in dark, iy as the vowel in she, aa as the vowel in dark, and ao as the first vowel in water., DL in audio outperforms shallow methods, when applied to raw data. Here we only have 5 features extracted from the raw data (its audio)

### POKER-HAND

**Tags**: Synthetic or Untraceable, Tabular

**#Samples**: 1025009    **#Features**: 9    **Year**: 2007

**Comments**: A task of classifying a poker hand based on it's content. One line non-ML solution exists, does not correspond to a real-world ML problem.

### POL

**Tags**: Tabular, Timesplit Needed

**#Samples**: 10082    **#Features**: 26    **Year**: 1995

**Comments**: The data describes a telecommunication problem, no further information is available.

### PROFB

**Tags**: Tabular, Timesplit Needed

**#Samples**: 672    **#Features**: 10    **Year**: 1992

**Comments**: Dataset describing professional football games. The task is to predict whether the favoured team was playing home.

### QSAR-BIODEG

**Tags**: HetE

**#Samples**: 155    **#Features**: 42    **Year**: 2013

**Comments**: The QSAR biodegradation dataset was built by the Milano Chemometrics and QSAR Research Group. Nowadays, a different approach based on graph neural networks is taken towards the task of predicting the characteristics of molecules, which is why this is not really a realistic use-case for tabular DL

### RL

**Tags**: Synthetic or Untraceable, Tabular

**#Samples**: 4970    **#Features**: 12    **Year**: 2018

**Comments**: Unknown real-life problem. Small, not many features, No canonical split. Retrieval methods such as TabR achieve large performance gains, which could signal that there is leakage in the data.

### ROAD-SAFETY

**Tags**: Tabular, Timesplit Needed, Timesplit Possible

**#Samples**: 111762    **#Features**: 32    **Year**: 2015

**Comments**: The data describes road accidents in Great Britain from 1979 to 2015. The task is to predict sex of a driver based on information about an accident. Retrieval methods such as TabR achieve large performance gains, which could signal that there is leakage in the data.

## SOCMOB

**Tags**: Tabular, Timesplit Needed

**#Samples**: 1156    **#Features**: 6    **Year**: 1973

**Comments**: An instance represents the number of sons that have a certain job A given the father has the job B (additionally conditioned on race and family structure). Just statistic data, not a real task

## SPLICE

**Tags**: Raw

**#Samples**: 3190    **#Features**: 61    **Year**: 1992

**Comments**: The task is to classify parts of genom as splice regions. The features are just a subsequence of DNA, more of an NLP task

## SULFUR

**Tags**: Leak, Tabular

**#Samples**: 10081    **#Features**: 6    **Year**: 2007

**Comments**: Leakage. In this dataset, there originally were 2 closely related target variables: H2S concentration and SO2 concentration. However, the version used in the aforementioned tabular benchmarks contains one of these target variables as a feature. According to the observed feature importance, the new feature is much more informative about the target variable than any of the old ones: the original features only describe the outputs of the physical sensors, while the new one already uses the knowledge about the chemical makeup of the gas. Due to the described problems, which stem from the accidental error in the data preparation, the current version of this dataset does not seem close to the intent of the original dataset authors.

## SUPERCONDUCT

**Tags**: Tabular

**#Samples**: 21263    **#Features**: 79    **Year**: 2021

**Comments**: This dataset presents information about superconductors, with a task of predicting critical temperature.

## VEHICLE

**Tags**: HetE

**#Samples**: 846    **#Features**: 19    **Year**: 1987

**Comments**: This dataset was created from the vehicle silhouettes in 1987, the task is to classify a car class by its silhouette.

## VISUALIZING_SOIL

**Tags**: Leak, Tabular

**#Samples**: 8641    **#Features**: 4    **Year**: 1993

**Comments**: Leakage. This dataset describes a series of measurements of soil resistivity taken on a grid. The original intended target variable was the resistivity of the soil, however, it wasn't the first

variable, and the technical variable #1 became the target variable in the later versions of this dataset on OpenML and in the tabular benchmarks. This makes the task absurd and trivial, as a simple if between two linear transforms of two different other features in the dataset performs on par with the best algorithm mentioned in the TabR paper, beating 4 others.

### WINE

**Tags**: Tabular

**#Samples**: 2554    **#Features**: 11    **Year**: 2009

**Comments**: This dataset was published by Cortez et al. in 2009, and it contains the chemical properties of different wines. The task is to predict the quality of wine.

### WINE_QUALITY

**Tags**: Tabular

**#Samples**: 6497    **#Features**: 11    **Year**: 2009

**Comments**: This dataset was published by Cortez et al. in 2009, and it contains chemical properties of different wines. The task is to predict the quality of wine.

### YPROP_4_1

**Tags**: HomE

**#Samples**: 8885    **#Features**: 62    **Year**: 2003

**Comments**: This dataset describes a series of chemical formulas, with a task of predicting one attribute of a molecule based on many others. The task would be better solved by graph DL methods.

