# OpenReview forum: "TabReD: Analyzing Pitfalls and Filling the Gaps in Tabular Deep Learning Benchmarks"
_ICLR.cc/2025/Conference — ICLR 2025 Spotlight_

### Official Review · Reviewer_s8Za · 2024-10-28

**Soundness:** 2
**Presentation:** 3
**Contribution:** 1
**Rating:** 3
**Confidence:** 4

**Summary:**

The paper discusses two major issues within the area of tabular data: time-shifting and rich feature datasets, both of which are often underrepresented. It mentions 100 datasets from academic benchmarks to highlight these problems. The author argues that both issues can lead to decreased performance and remain largely unexplored.

To address these challenges, the author introduces a new framework called TabReD, designed for conducting experiments in this specific context. The experiments utilize eight datasets that represent real-world data. Additionally, the paper compares TabReD to a framework developed by Gorishniy. The results indicate that TabReD can effectively serve as a framework for these experiments.

Furthermore, the findings reveal that the results of the time-shifting experiments exhibit higher variance.

**Strengths:**

The paper shows experiments on 8 different databases representing real-world scenarios. The experiment results are thoroughly presented and explored, The paper provides up-to-date comparisons with other major papers.
The code provided are very useful.

**Weaknesses:**

Some terms are missing, and the paper is not self-contained. Take data leakage, for example. The writer cites another paper, which I believe is the writer's paper, but does not give a glimpse of the definition.

One of the main arguments is data leakage; however, only 11 (10%) were detected among 100 datasets, which is not significant.

The main issues are domain shifting and time shifting. Both areas are not new and have been explored very well. Domain shifting is also called an out-of-domain or out-of-distribution Problem. This problem is not new and is well-known within academic research.

For the second problem, time shifting split is a vague problem. There are reasons why random split is widely used. The main reason is that a random split is expected to have a similar distribution, unlike a shifting split. The result of a model trained in a time-shifting split is expected to vary between experiments; this is just because it came from a different distribution.

TabRed, we do not need to reinvent the wheel for each problem. Gorishniy's framework is more than enough for the experiment. If your focus is on the above problems, then you can just adjust the Dataloader class from Pytorch.

While there were some models inside, the comparison was made only to one single framework. The related work mentioned more than one. Some experiments may be helped to pursue reader the importance of TabReD.

The paper has many details, but they are written within the appendix. The main pages lack technical details. To understand the paper completely, the 10 main pages are not sufficient. This is actually a very long paper.

**Questions:**

What is the definition of data leakage, and why is it a problem?
Similar distribution with training and testing is not necessarily called data leakage.

---

> ### Author Response · Authors · 2024-11-20
> **Response (Part 1/2)**
>
> Thank you for the review. We address your concerns and reiterate relevant paper contribution below.
>
> ---
> ### **Clarifying the core contribution of the paper**
>
> Distinction between "Framework" and "Benchmark":
>
> > ...the author introduces a new **framework** called TabReD...
>
> > TabRed, we do not need to **reinvent the wheel** for each problem. Gorishniy's **framework** is more than enough for the experiment. If your focus is on the above problems, then **you can just adjust the Dataloader class** from Pytorch
>
> It is important to clarify that our main contribution is in **introducing new datasets** from real-world tabular problems with temporally shifting data and extensive feature-engineering that are often encountered in industry but rarely available in existing benchmarks. **We do not introduce new frameworks** in the submission and use an established protocol/framework (by [Gorishniy et. al.](https://arxiv.org/abs/2307.14338)) to evaluate popular techniques.
>
> We also **actively discourage viewing this benchmark as a replacement for existing tabular benchmarks** in section 4.1. We provide this benchmark for testing methods in previously underrepresented conditions and do not discourage testing on random splits and previously available datasets.
>
> ---
>
> Core contribution - the need for representative datasets
>
> > The main issues are domain shifting and time shifting. Both areas are not new and have been explored very well. Domain shifting is also called an out-of-domain or out-of-distribution Problem. This problem is not new and is well-known within academic research.
>
> We discuss relevant prior work on temporal shift and domain shift in both general ML ([Yao, Huaxiu, et al.](https://arxiv.org/abs/2211.14238)), tabular DL ([Sergey Kolesnikov et al.](https://arxiv.org/abs/2312.01792), [Josh Gardner et al.](https://arxiv.org/abs/2312.07577)) and specific applications ([Roger M. Stein](https://www.rogermstein.com/wp-content/uploads/BenchmarkingDefaultPredictionModels_TR030124.pdf), [Chip Huyen](https://books.google.ru/books?id=EThwEAAAQBAJ), [Daniel Herman et al.](https://www.kaggle.com/competitions/home-credit-credit-risk-model-stability)) in the related work section. One of the key insights from this line of work is precisely that evaluating methods on identically distributed data may not predict real-world performance when temporal shift is present, which brings us to the second point
>
> > For the second problem, time shifting split is a vague problem. There are reasons why random split is widely used. The main reason is that a random split is expected to have a similar distribution, unlike a shifting split. The result of a model trained in a time-shifting split is expected to vary between experiments; this is just because it came from a different distribution.
>
> We cannot agree with the reviewer's claim. First, **temporal Shift is a concrete and common phenomenon**, even amongst existing datasets (see numerous sources we cite above, in related work and in section 3 discussion). Random splits may not be indicative of real-world performance in deployment scenarios where data naturally evolves over time. As we demonstrate experimentally in section 5.4 methods' relative performance can change significantly when evaluated on time-based splits that better match deployment conditions. This is a systematic difference, not merely higher variance due to a different distribution.
>
> Furthermore, the **temporal shift is not the sole characteristic of the new datasets**. In addition to temporal splits, TabReD datasets have rich feature sets (median 261 features vs 13-23 in existing benchmarks, see [Table 1]). This reflects substantial industrial investment in feature engineering, as detailed in Section 4. This characteristic of TabReD datasets also affects existing tabular DL techniques differently (see the new results on this in the general response).
>
> Both temporal-shifts and extensive feature engineering are characteristics common in industrial applications that are underrepresented in the existing benchmarks. We cover **this specific scenario** with TabReD. As we state in section 4.1, TabReD serves as a **further evaluation** for demonstrating industrial usefulness of novel techniques in tabular DL. We **do not discards existing IID benchmarks**, which are still important and useful.

---

> ### Author Response · Authors · 2024-11-20
> **Response (Part 2/2)**
>
> > Q: What is the definition of data leakage, and why is it a problem? Similar distribution with training and testing is not necessarily called data leakage.
>
> Regarding data leakage - we use this term for datasets that include information that's unavailable during model deployment.  This is a common definition ([Chip Huyen, chapter on feature engineering](https://books.google.ru/books?id=EThwEAAAQBAJ), [Kaggle ML Course](https://www.kaggle.com/code/alexisbcook/data-leakage
> ), [Sayash Kapoor et al.](https://pmc.ncbi.nlm.nih.gov/articles/PMC10499856/)). We will add a note on this to the paragraph about leaks in section 3 shortly. In some datasets, one of original target variables is added to the features (e.g. SGEMM_GPU_kernel_performance, sulfur), while some datasets have an answer to one example as a feature in other, and originally used non-random split to avoid leakage, which was changed to random in later literature (eg. Facebook Comments Volume, electricity).
>
> > One of the main arguments is data leakage; however, only 11 (10%) were detected among 100 datasets, which is not significant.
>
> As we discuss above, pointing out data-leakage issues is not the main argument in our paper. Nevertheless, it is an important observation we make from dataset analysis. We do not agree that this is not significant, **11**% can noticeably affect the evaluation results.
>
> ---
> ### **Experiments and Technical Details**
>
> > While there were some models inside, the comparison was made only to one single framework. The related work mentioned more than one. Some experiments may be helped to pursue reader the importance of TabReD.
>
> We believe that the breadth of comparison in Figure 1 is not a priority, as it is addressed by:
>
> 1. Consistency of our results obtained on the Gorishniy benchmark with the results from the tabular DL literature. For example, retrieval methods (TabR, ModernNCA), embeddings and ensembles show consistent performance across hundreds of datasets in existing benchmarks (the largest one to date is [Han-Jia Ye et al.](https://arxiv.org/abs/2407.00956), it covers many datasets from [Léo Grinsztajn et al.](https://arxiv.org/abs/2207.08815) and Tabzilla). Strenghts of the long-training with augmentations is also replicated on a broader set of datasets ([Kyungeun Lee et al.](https://arxiv.org/abs/2405.07414)).
> 2. Extensive testing on TabReD under controlled conditions:
>     - Analysis of temporal vs random splits impact on model ranking in Section 5.4
>     - Analysis of the feature-rich vs "typical academic" feature-engineering regimes on model ranking in the rebuttal experiments (see general responsе), we'll update the pdf with these additional results shortly.
>
> Experimental results demonstrate that both temporal drift and extensive feature engineering present in industrial data lead to qualitatively different model behaviors compared to existing benchmarks.
>
> > The paper has many details, but they are written within the appendix. The main pages lack technical details. To understand the paper completely, the 10 main pages are not sufficient. This is actually a very long paper.
>
> We believe that the size (9.5 pages) of the paper is justified by many figures and detailed discussion of the new benchmark position and motivation. What specific missing details are you referring to?

---

> ### Comment · Reviewer_s8Za · 2024-11-25
>
> Thank you to the authors for their hard work; I truly appreciate the update.
>
> Data Leakage is a common term, but it has several meanings. For example, in security, Data leakage could mean a data breach. [1]. The paper mentioned was written in 2016.
>
> ``As we discussed above, pointing out data-leakage issues is not the main argument in our paper. Nevertheless, it is an important observation we make from dataset analysis. We do not agree that this is not significant, 11% can noticeably affect the evaluation results.``
> I agree that this may not be the primary focus of the paper. Still, I believe it serves as an essential foundation for justifying it, especially as it challenges other available benchmarks that use the 100 datasets.
>
>
> ``We believe that the size (9.5 pages) of the paper is justified by many figures and detailed discussion of the new benchmark position and motivation. What specific missing details are you referring to?``
>
> In my understanding, a key objective of this paper, as highlighted in the abstract, is: *We demonstrate that evaluation on time-based data splits leads to different methods ranking, compared to evaluation on random splits, which are common in academic benchmarks. F.* However, we could not find specific technical details on how this is implemented in Section 5.1. Instead, it is suggested to refer to the appendix for further information.
>
> >For extended description of our experimentalsetup including data preprocessing, dataset statistics, statistical testing procedures and exact tuninghyperparameter spaces, see Appendix C. Below, we describe the techniques we evaluate on TabReD.
>
> I am not addressing the concept of shifting distribution, as it is indeed a valid phenomenon. Instead, I am focusing on the time split method utilized by the authors. The authors did not demonstrate that implementing a time shift split leads to a distribution shift. Rather, they presented performance variations, which are anticipated when a non-random shuffle is applied. I believe time shifting leads to temporally static, which tends to overfit [2]. And I believe that is the reason for the big variation in performance.
>
> [1] A survey on data leakage prevention systems. 2016
> [2] Improving performance of spatio-temporal machine learning models using forward feature selection and target-oriented validation

---

> > ### Author Response · Authors · 2024-11-25
> > **Response to Response (1/2)**
> >
> > Thank you for addressing our response.
> >
> > > I agree that this may not be the primary focus of the paper. Still, I believe it serves as an essential foundation for justifying it, especially as it challenges other available benchmarks that use the 100 datasets.
> >
> > We do not want to challenge other available benchmarks. The comment about 11 data-leakage containing datasets is aimed at marginal improvement of existing benchmarks, not their deprecation or reduced usage. The point of our paper is that other benchmarks are leaving uncovered two settings: time-shifting data and rich feature sets. We believe that on data outside these settings, current benchmarks give an informative estimation of practical usefulness. We had an extended discussion of this topic with reviewer R-9YJo and modified our paper to much better reflect the validity of prior benchmark for their intended setting.
> >
> > We do, however, believe that our modified setting could lead to different methods ranking. To demonstrate this, we chose a simpler way of data presentation. In the table below, you can see percentage improvements over MLP averaged over our datasets. Random splits row refers to evaluation on the same data points where train, val and test sets were sampled randomly instead of temporally split. 20 features row refers to evaluation on data where 20 features with highest feature importance scores on XGBoost were selected.
> >
> > ||MLP(ens.)|MLP(aug.rec.)|MLP-PLR|MLP-PLR(ens.)|TabR|XGBoost|
> > |----|----------|---------------|-------|--------------|---|---|
> > |TabReD|0.67|-1.98|0.61|1.28|-2.78|0.91|
> > |TabReD (Random split)|0.28|-0.47|0.47|0.96|0.6|2.02|
> > |TabReD (Top 20 features)|0.52|-0.13|0.69|0.98|1.09|0.97|
> >
> > We would like to point out several key insights from this table.
> >
> > 1. Ensembles of MLP-PLR overperform XGBoost on canonical TabReD. This distinction disappears when changing split to a random one and changes to a tie when selecting only 20 features. Also, the gap between MLP-PLR and XGBoost reduces greatly on canonical TabReD.
> > 2. TabR becomes much more useful on 20 features mode, and a little more beneficial on random samples.
> > 3. The harmfulness of augmentation techniques reduces greatly on both random split and 20 features evaluations.
> > 4. In the updated submission pdf, we emphasize the importance of both timeshifting and feature-rich settings. We updated the paper to better reflect the equal importance of these two changes. In the abstract, we now refer to studying the effects of both time-splits and feature richness.

---

> ### Author Response · Authors · 2024-11-25
> **Response to Response (2/2)**
>
> > The authors did not demonstrate that implementing a time shift split leads to a distribution shift.
>
> The table above demonstrates the change in relative performance between methods due to timeshift and richer features. While we believe the change in relative performance gives motivation to use temporal splits, which are closer to practice for applications described in this paper, we would like to closely focus on why we believe they are explained by distribution shift.
>
> First, the explanation from your [2] citation does not contradict the existence of distribution shift in time. The differences in some features that arise when applying time split could be explained by temporal distributional shift. Second, "Wilds: A Benchmark of in-the-Wild Distribution Shifts", (by [Pang Wei Koh and Shiori Sagawa et al.](https://arxiv.org/pdf/2012.07421)), define distribution shifts as a situation where "the training distribution differs from the test distribution". We would like to highlight that on our datasets, not only the distribution of some features changes from train to test, but these features are important for the models we used. We measure Wasserstein distance between train and test distributions of the target variable Y for the random and temporal data splits and an average Wasserstein distance between train and test for top-20 most important features. The distances are summarized in the following table:
>
> | Dataset | Temporal Split Y distance | Temporal Split Features Distance | Random Split Y distance | Random Split Features Distance |
> |----|----|----|----|---|
> | Sberbank Housing | 8.79e-02 | 9.44e-02 | 9.68e-03 | 2.22e-02 |
> | Ecom Offers | 1.31e-01 | 2.82e-01 | 2.65e-04 | 7.30e-03 |
> | Maps Routing | 5.58e-02 | 1.01e-01 | 3.16e-03 | 6.61e-03 |
> | Homesite Insurance | 3.74e-03 | 5.52e-02 | 4.23e-03 | 6.32e-03 |
> | Cooking Time | 6.90e-02 | 7.06e-02 | 4.38e-03 | 8.07e-03 |
> | HomeCredit Default | 9.18e-03 | 1.53e-01 | 1.07e-03 | 4.19e-03 |
> | Delivery ETA | 6.44e-02 | 1.68e-01 | 9.96e-03 | 7.67e-03 |
> | Weather | 5.60e-01 | 1.29e-01 | 3.08e-02 | 6.13e-03 |
>
> For all datasets both target and top features distributions are further apart for the temporal data splits, indicating the presence of shift. Similar notions of distribution shifts were used in [Gardner et al.](https://arxiv.org/abs/2312.07577) for indicating target shifts for example (they used a simpler, absolute difference between average targets for train and test sets)
>
>
> > However, we could not find specific technical details on how this is implemented in Section 5.1. Instead, it is suggested to refer to the appendix for further information.
>
> We changed the experimental setting to a comparison between canonical TabReD and the same data but split randomly on train/val/test parts for simplicity and included these details in the section (instead of using different sliding window splits in this section). We also updated section 5.4 from only focusing on the influence of time shift to also include the experiment on the influence of richer feature sets.
>
> To sum up this response:
> 1. We actively avoid challenging previous benchmarks, and think they do a great job with demonstrating behaviour on non-timeshifting and non-feature rich data.
> 2. We demonstrate that method ranking changes when we "turn off" the two novel conditions described in this paper. This demonstrates that random-split evaluation could lead to wrong conclusions for applications on our data, since they perform on data in the future relative to the one on which they were trained.
> 3. Using the definition of distribution shift from "Wilds: A Benchmark of in-the-Wild Distribution Shifts", we demonstrated that in our datasets, there is a distribution shift between train and test subsets.
> 4. We changed the paper to be more intuitively understandable, study the effects of both of our new settings instead of just one, and create less impression of deprecating previous benchmarks.
>
> Since the discussion period is almost over, could you reiterate the concerns about our paper that you still have after this response? We would greatly appreciate it and it would help us improve our work.

---

> > ### Author Response · Authors · 2024-11-28
> > **Invitation to discussion**
> >
> > Since the PDF change deadline is soon, we updated our PDF to include the above demonstration of distribution shift in the appendix A.
> >
> > Since the discussion period is extended until December 3rd we are open to discuss our response if you have any concerns left.

---

### Official Review · Reviewer_9YJo · 2024-10-30

**Soundness:** 2
**Presentation:** 3
**Contribution:** 3
**Rating:** 8
**Confidence:** 4

**Summary:**

Motivated by the current landscape of academic benchmarks for tabular data, this work aimed to explore the limits of such benchmarks and provide an alternative. In detail, the paper proposes eight new datasets as part of a benchmark called TabReD. Moreover, the paper analyzes several prior benchmarks for tabular data and evaluates deep learning vs. classical machine learning methods on TabReD.

The contributions of the paper are:
* Analysing prior tabular benchmarks that focused on IID* tasks.
* TabReD with datasets originating from Kaggle and newly sourced from the industry, focus on non-IID with a temporal shift.
* Evaluating classical machine learning, recent deep learning methods, training methodologies, and MLP ensembles on TabReD.
* Highlight the difference between the conclusion from the result on TabReD and one from one prior benchmark.
* Highlight the difference between time-based (no IID assumption) and random-based (IID assumption) splits.

\* In my wording, I use IID (or iid) to refer to machine learning tasks applied to data from independent and identically distributed random variables. Likewise, this makes the ML task predict by interpolation instead of extrapolation.

**Strengths:**

The paper tackles the important problem of tabular benchmarking and its limits in academic practice. Tabular predictive modeling for non-iid tasks (such as under a temporal shift) is rarely correctly discussed and largely overlooked in our field. Thus, I see this work as eye-opening to many academics and hopefully inspiring better benchmarking practice in the future. Furthermore, providing reliable and usable benchmarks for this problem type is helpful for future researchers, as well as helping us to determine trends that generalize across task settings (e.g., PLR working well for non-iid and iid data).

The related work section is well-written and sufficiently exhaustive to provide the correct context. It fits the framing provided in this work.

Given the experimental design and datasets, the results look reasonable and, from the code, reproducible. The observation that traditional boosting outperforms recent tabular deep learning approaches matches prior (big) benchmarks. Moreover, the power of ensembling also fits into this picture. Section 5.3 is a highlight among the results and an exciting avenue for meta-studies. Section 5.4 is interesting and useful empirical evidence but not surprising or unexpected.

The experimental design for training, tuning, and running baselines is mostly in line with prior work on deep learning vs. classical machine learning. Thus, it seems to be accepted by the community as sufficient. In other words, while there is a lot of room for improvements (e.g., better search spaces or non-standard preprocessing for filling NaNs), this is, to me, not the core contribution of this work, and improving such points would likely not change the conclusions presented in the work.

**Weaknesses:**

I detail the weaknesses of this work in subsections, starting with the biggest weakness.

## Discussion and Framing of non-IID vs. IID Machine Learning

Prior benchmarks for tabular data that focus on classical machine learning vs. deep learning assume that the data is IID. Thus, the framing of this work seems problematic to me, as it (at least) implies that prior tabular benchmarks desire to determine how good tabular models are for non-IID data.

I see where one might get this impression from in the prior classical machine learning vs. deep learning benchmarks, as they have little formalized assumptions about their data. Moreover, this is rarely mentioned or specified by method developers that use such benchmarks either. Nevertheless, from my perspective, the core assumption for many algorithms and research in this domain is still IID data.

Consequently, this work positions itself as a solution to an ill-specified problem. If I am benchmarking for IID data, I might not care about performance for non-IID tasks such as the temporal shift (i.e., I do not see this as a gap that needs to be filled). In general, the need for time splits is a clear indicator that we might need to change methodology away from pure tabular IID-based machine learning and switch to (multivariate) forecasting with (many) exogenous variables.

The paper crucially lacks a discussion about this perspective of IID vs. non-IID. The authors seem motivated by a real-world application that could have been better solved without pure-tabular IID-based models. To be more precise, the missing discussion is about which machine learning tasks we want to solve. Prior tabular benchmarks evaluate which models are good at interpolation, while this work wants to evaluate which models are good at extrapolation. But somehow, this distinction has not happened, confusing the results and conclusion.

In summary, this paper lacks a discussion about IID vs. non-IID and should likely even include forecasting algorithms as a baseline (especially for the regression tasks; for classification, it gets more complicated). Moreover, the title and framing of the abstract/introduction could be adjusted such that this work is more about a new, specific benchmark than an alternative or extension of prior benchmarks. The results would fit such a framing even better.

## Related Benchmarks

While the work correctly identifies the big benchmarks for deep learning vs. classical machine learning (minus too recently published works), it fails to include all big classical machine learning benchmarks (for IID data). This is problematic as several of the claims and results in this paper seem independent of the deep learning debate.

To illustrate, the paper states "the landscape of existing tabular machine learning benchmarks compared to TabReD" in Table 1, which is categorically wrong. It would be more correct to claim "the landscape of existing tabular deep learning benchmarks compared to TabReD".

For general tabular benchmarking, sets like the AutoML Benchmark Suite [1], or TabRepo [2] are missing. While most of these also exhibit some of the problems detailed in this work, the AutoML Benchmark Suite is better curated than deep learning vs. classical machine learning benchmarks. In comparison, TabRepo shows how the correct search spaces, tuning, meta-learning, and ensembling can boost performance for tabular IID-based modeling.

Finally, the field of time series forecasting offers many more benchmarks for predictive modeling with non-IID data and (slight) temporal shifts. For example, see the benchmark and datasets used in AutoML or foundation models for time series [3, 4].

This criticism is mostly related to framing and careful wording. The author should ensure that their work is clearly differentiated from such prior work or incorporate it into their analysis.

## Kaggle Data

The paper presents data from Kaggle as the "holy grail" for real-world data and benchmarks. This is an overly positive framing, given that Kaggle has many problems with its data and also hosts synthetic competitions. Specifically, the paragraph about Kaggle in the related work section (Line 113) lacks any critical reflection of data from Kaggle. To give some examples, data from competitions on Kaggle can be duplicates, sourced from UCI or OpenML, purely synthetic data, badly preprocessed, or uploaded with a data leak (see [5] for a related and extended discussion).

Furthermore, sourcing data from Kaggle competitions and employing the top-performing preprocessing pipelines has two critical problems not discussed in this paper: 1) the pipeline is selected on test performance (leaderboard score) and thus leaking or overfit; 2) we do not have the target labels for the test data.

\2) is specifically bad if, for example, the data only exhibits a crucial distribution shift in the test (or a temporal shift). Then, determining how good a model is for the training data is not representative of the real-world task that is to be solved by the competition. Likewise, the pipeline selected for preprocessing might not be representative of the training data or comparing models on this data.



# References
1. Gijsbers, Pieter, et al. "Amlb: an automl benchmark." Journal of Machine Learning Research 25.101 (2024): 1-65.
2. Salinas, David, and Nick Erickson. "TabRepo: A Large Scale Repository of Tabular Model Evaluations and its AutoML Applications." arXiv preprint arXiv:2311.02971 (2023).
3. Shchur, Oleksandr, et al. "AutoGluon–TimeSeries: AutoML for probabilistic time series forecasting." International Conference on Automated Machine Learning. PMLR, 2023.
4. Ansari, Abdul Fatir, et al. "Chronos: Learning the language of time series." arXiv preprint arXiv:2403.07815 (2024).
5. Tschalzev, Andrej, et al. "A Data-Centric Perspective on Evaluating Machine Learning Models for Tabular Data." arXiv preprint arXiv:2407.02112 (2024).

**Questions:**

* Why does the paper only show 36 instead of 176 datasets for TabZilla?
* Several interesting aspects could be added to the overview of the new datasets: What year are the new datasets from? What task (classification, regression, forecasting) are they intended for? Which metric is used to measure performance for these datasets? What are the features about?
* The appendix does not provide enough detailed information about the new datasets and how they were preprocessed. Moreover, it is unclear which feature pipeline was chosen for Kaggle datasets. Such information are important and would be needed for something like dataset cards.
* Why is there no ensemble of classical machine learning methods? And what kind of ensemble is it for the MLP?
* What is the variance of the scores across seeds? Why is no backtesting for time splits considered (akin to cross-validation)?
* I seem like the plots in Figure 2 are needlessly complex to read for the point that is made. Maybe replace the metric with the rank to make the message clearer? (also note the arrows for higher/lower is better are pointing in the wrong directions right now)

---

> ### Author Response · Authors · 2024-11-20
> **Response (Part 1/3)**
>
> We sincerely thank you for your review! We think it raised many important points and will help us make our paper better. Below, we address your concerns about our submission.
>
> ---
> ## On I.I.D. vs non-I.I.D. setting:
>
> Thank you for your comments about I.I.D. vs non-I.I.D. settings. We will add more clarifications on the positioning of our work in subsection 4.1.
>
> We agree that benchmarks that cover only I.I.D. setting are valid and useful. We did not intend to give an impression that we treat the choice to limit a benchmark to only I.I.D. setting as a disadvantage.
>
> In non-I.I.D. setting, one could choose to follow one of the two possible approaches: the first is to develop a method that explicitly takes timeshift into account and tries to adjust the predictions with the flow of time, and the second is to just test different approaches that work for I.I.D. data and use them in non-I.I.D. conditions. In the following part of our response, we will demonstrate that on the kind of datasets this paper focuses on, timeshift-agnostic methods are usually used instead of the forecasting methods.
>
> **Discussion of timeshift-agnostic vs forecasting methods**
>
> We believe there is a separation between forecasting tasks and our non-I.I.D. tabular prediction tasks. For our datasets, we specifically chose datasets with rich features, where the time feature does not overwhelm the importance of other features. In this regard, our task is different from time-series tasks. When looking at the discussion section on Kaggle, we notice that none of the competitions from which we source our data have a competitive solution based on forecasting. Here are the details:
> - On Sberbank Housing dataset, the only mention of forecasting is for feature engineering ([Forum link](https://www.kaggle.com/competitions/sberbank-russian-housing-market/discussion/35631)). Even in that context, forecasting was not successful and failed to improve performance of the downstream model
> - On Homesite Insurance dataset, the only mention of utilizing time-series is again for feature engineering, and again it only gives an insignificant boost to the score of 2e-4 ([Forum link](https://www.kaggle.com/competitions/homesite-quote-conversion/discussion/18826)). All top decisions mentioned in discussion utilize shift-agnostic models such as XGBoost.
> - On the E-Commerse dataset, time and forecasting is not mentioned at all, and the top solution focuses on feature engineering approaches ([Forum link](https://www.kaggle.com/competitions/acquire-valued-shoppers-challenge/discussion/9756)).
> - On HomeCredit dataset, the mistake in metric choice by hosts resulted in some conditions using unrealistic hacks to win. Nevertheless, in the discussion process, competitors agreed, than when evaluation by pure AUC, feature engineering is more critical ([Forum link](https://www.kaggle.com/competitions/home-credit-credit-risk-model-stability/discussion/497167)).
>
> For our newly introduced datasets, no forecasting experiments have proved useful in production, and simple shift-agnostic tabular ML models have proven to be useful in real-world applications scenarios.
>
> Overall, we believe tabular DL/ML and forecasting/time-series are different domains, and we chose to focus on OOD situation in tabular data. In that sense, our paper is closer to Wild-Time ([Huaxiu Yao et al.](https://arxiv.org/abs/2211.14238)), which also focuses on timeshifting data, albeit in other domains. In that paper, forecasting methods are not used, instead the authors utilize general increased robustness methods, and in this work, we benchmark robustness methods like DFR and CORAL too. Interestingly, in a related field of distributional shift robustness, there are questions about the effectiveness of methods trying to mitigate non-I.I.D. data influence on model’s performance (e.g. In Search of Lost Domain Generalization, [Ishaan Gulrajani et al](https://arxiv.org/pdf/2007.01434))

---

> ### Author Response · Authors · 2024-11-20
> **Response (Part 2/3)**
>
> **The role of TabReD benchmark**
>
> According to our analysis, more than half of 100 datasets from previous benchmarks come from domains that involve non-I.I.D. setting in real-world deployment. For example, in weather prediction, in applications, usually the task is to predict weather in the future relative to the data available for training. Yet, we do believe that an I.I.D. split of the weather dataset could be useful for comparison of a tabular model’s performance on I.I.D. data. However, we think that the fact that a majority of tabular datasets from the popular benchmarks come from tasks that essentially assume non-I.I.D setting in deployment indicates that researchers are interested in how tabular models work on both I.I.D. data and non-I.I.D. data, however previous benchmarks only focus on one of these fields. In that sense, there is a gap in possibilities of evaluation on current benchmarks, and that is the gap that we meant to address with this paper.
>
> > the title and framing of the abstract/introduction could be adjusted such that this work is more about a new, specific benchmark than an alternative or extension of prior benchmarks.
>
> We **actively discourage** future researchers from treating this benchmark as an alternative to prior benchmarks, describing our work in 4.1 as useful for “further evaluation” of tabular DL methods. However, we believe TabReD could be used as an extension of prior benchmarks. If a new tabular method is developed, it would also be interesting to see how well does it perform on new conditions facilitated by TabReD: time-based split and richer feature sets. If deemed interesting by a researcher, timeshift’s effect could be isolated by also testing a model on random splits of TabReD, which we also provide.
>
> The way in which a model interacts with non-I.I.D. and feature-rich regimes would provide additional information on the model’s limitations. We also do not believe that failure on non-I.I.D. data should negatively influence model’s image in I.I.D. applications, and do not think a negative result on TabReD should prevent a method’s publication, but rather bring more understanding about limits of applicability of the method.
>
> To sum up the whole discussion, our work is not really about time-series and forecasting methods, but about a different application scenario for tabular methods. We believe our benchmark is closer to testing OOD robustness of different methods, like in (Wild-Time: A Benchmark of in-the-Wild Distribution Shift over Time, by Huaxiu Yao et al., https://arxiv.org/abs/2211.14238), but we also cover a second uncovered condition: in this benchmark, there are much more features than in previous ones, and we offer random split versions of TabReD to make it possible to isolate this effect from the time-split OOD influence. We think that purely I.I.D. setting is still valid, and do not wish to disparage it. As a result of this discussion, we will add more on the place of I.I.D. vs non-I.I.D. in our paper to the Role and Limitations section. We would also like to draw your attention to the feature-rich character of our data, which marks another shift from previous tabular DL benchmarks (and new results highlighting it in the general response).
>
> ---
> ## Related Benchmarks
>
> We agree that we should have chosen a different wording when describing related benchmarks. We will replace our general terms to refer specifically to tabular deep learning benchmarks.
>
> > Finally, the field of time series forecasting offers many more benchmarks for predictive modeling with non-IID data and (slight) temporal shifts.
>
> We would like to differentiate the field of TabReD from the field of time-series. This benchmark explicitly focuses on feature-rich scenarios, in which default tabular data methods perform better than time-series approaches (as we show in Kaggle challenges above).

---

> ### Author Response · Authors · 2024-11-20
> **Response (Part 3/3)**
>
> ---
> ## Kaggle Data
>
> > The paper presents data from Kaggle as the "holy grail" for real-world data and benchmarks.
>
> > Specifically, the paragraph about Kaggle in the related work section (Line 113) lacks any critical reflection of data from Kaggle.
>
> We only meant that there are more good datasets on Kaggle than in other sources. We do agree that a general dataset from Kaggle suffers from many issues, and conducted a thorough checking for which datasets we select. We will add a note that we do not refer to all datasets, but only to the selected best ones.
>
> > To give some examples, data from competitions on Kaggle can be duplicates, sourced from UCI or OpenML, purely synthetic data, badly preprocessed, or uploaded with a data leak
>
> The discussion section is often useful in identifying these problems. All our selected datasets are free from these issues. The selection criterions are discussed in part 4 and appendix D.
>
> > Furthermore, sourcing data from Kaggle competitions and employing the top-performing preprocessing pipelines has two critical problems not discussed in this paper: 1) the pipeline is selected on test performance (leaderboard score) and thus leaking or overfit;
>
> We choose only common-sense approaches to data engineering, avoiding some suspicious choices that may result in overfit (excluding datasets with known leakage, "magic" feature or other hacks). Moreover, no significant meta-overfitting was observed in a large scale study of Kaggle competitons ([Rebecca Roelofs et al.](https://dl.acm.org/doi/pdf/10.5555/3454287.3455110)) (modulo some outliers, which we do exclude). We also provide our preprocessing code in the repository, and the community is free to experiment with it to find out any suboptimalities in our choices.
>
> > 2 is specifically bad if, for example, the data only exhibits a crucial distribution shift in the test (or a temporal shift).
>
> Experiments in section 5.4 indicate that temporal shift is present in our provided splits (we've specifically chosen the datasets where the split is possible and the time-frames are realistic in terms of the underlying task)
>
> ### Questions:
>
> 1. In the paper introducing TabZilla (When Do Neural Nets Outperform Boosted Trees on Tabular Data?, by Duncan McElfresh et al., https://arxiv.org/pdf/2305.02997), the authors introduce “TabZilla Benchmark Suit” as a “collection of the 36 ‘hardest’ of the datasets we study”. We chose this version as an official version introduced in the paper.
> 2. We will add information about the time period from which the dataset comes in the appendix of our paper. All 4 of our newly introduced datasets focus on the regression task. We use RMSE for regression and AUC-ROC for classification. We cannot yet release some information about our datasets due to anonymity policy. However, we will provide the datasheet which we filled-out already.
> 3. We give links to discussion posts on Kaggle on which we based our data preprocessing in the datasheet. We also provide full preprocessing code for Kaggle datasets.
> 4. The scores for XGBoost, CatBoost, and LightGBM are in the **Table 1** bellow (first three datasets use AUC-ROC, the higher the better, the rest use RMSE, the lower the better). We did not provide ensemble scores originally, because the ensembles of boostings are usually not as effective as ensembles of NNs (Revisiting Deep Learning Models for Tabular Data, by Yury Gorishniy et al., https://arxiv.org/pdf/2106.11959)
> 5. We will upgrade the appendix to show std’s across seeds. Currently, we would like to draw your attention to subsection C.2, Table 5, which uses multiple comparison statistical test to obtain ranks of different methods.
> 6. We provide a more succinct version of the figure 2 in the general response.
>
> **Table 1**. Results for the ensembles of GBDTs:
> | | Homesite Insurance | Ecom Offers | HomeCredit Default | Sberbank Housing | Cooking Time | Delivery ETA | Maps Routing | Weather|
> |-|-|-|-|-|-|-|-|-|
> |XGBoost|0.9601|0.5763|0.867|0.2419|0.4823|0.5468|0.1616|1.4671|
> |XGBoost-ens.|0.9602|0.5917|0.8674|0.2416|0.4821|0.5463|0.1614|1.4629|
> |CatBoost|0.9606|0.5596|0.8621|0.2482|0.4823|0.5465|0.1619|1.4688|
> |CatBoost-ens.|0.9609|0.5067|0.8636|0.2473|0.482|0.5461|0.1615|1.4576|
> |LightGBM|0.9603|0.5758|0.8664|0.2468|0.4826|0.5468|0.1618|1.4625|
> |LightGBM-ens.|0.9604|0.5758|0.8667|0.2467|0.4825|0.5465|0.1616|1.4581|

---

> > ### Comment · Reviewer_9YJo · 2024-11-20
> > **Response to Response**
> >
> > Dear Authors,
> >
> > Thank you for your extensive response to my concerns and for addressing my concerns. I will raise my score once I review the new version of the manuscript that will be uploaded.
> >
> > I do not have any further concerns but will nevertheless comment on some points below.
> >
> >
> > ## Discussion of timeshift-agnostic vs forecasting methods / The role of TabReD benchmark
> >
> > Thank you for the great and interesting discussion (and for the links to the forum). This fully clarifies the gap you are filling for me that I might have missed before.
> >
> > I think that similar explicit (or repeated) arguments in the revised paper would greatly increase the work's impact. You could add a paragraph about "intended use" in the conclusion, summarizing the "Response (Part 2/3)." I believe this would directly prompt method developers to use the benchmark for these reasons.
> >
> > Interestingly, this might open the floor for more time-related feature engineering than algorithmic advances, which would be reflected for Kaggle but not academia (so far).
> >
> > ## Related Benchmarks
> >
> > Indeed, the feature-rich scenario is out of scope for time series. Even the best foundation models for time series cannot use extra features so far. I believe it would be worth highlighting this distinction in the related work as well.
> >
> > ## Kaggle Data
> >
> > My intention with the comment about Kaggle data was not to criticize your selected dataset but to put this paragraph in context, i.e., criticize its framing. Your defenses are nevertheless appreciated, and I mostly agree with them.
> >
> > Please note that Rebecca Roelofs et al. used mostly non-tabular tasks, and the results are, thus, limited in their representativeness of tabular data. [Tortman's shake-up analysis](https://www.kaggle.com/code/jtrotman/meta-kaggle-competition-shake-up) might be more insightful for tabular data.
> >
> > ## Questions
> >
> > 1. Indeed, thank you for the clarification. I have seen so many papers that associated all 176 datasets with TabZilla that I did not remember this detail.
> > 4. Thank you for the additional results. However, I must disagree with the sentiment that ensembles of boosting are usually not as effective as ensembles of NN. The ensembles by Yury Gorishniy et al. lack performance due to their type (unweighted averages of random groups of models). One can clearly see in TabRepo, Table 1, that (weighted post-hoc) ensembles of (tuned) GBDTs significantly improve performance.

---

> > > ### Author Response · Authors · 2024-11-21
> > > **PDF Updates**
> > >
> > > We modified our manuscript in reponse to your concerns.
> > >
> > > 1. We adjust the wording to specifically refer to tabular benchmarks dedicated to comparison between DL and ML models.
> > > 2. We added a sentence to the introduction that explains that current benchmarks allow experiments in I.I.D. conditions, but have a gap with non-I.I.D. conditions common in applications.
> > > 3. We rewrote subsection 4.1. We believe the new version better reflects that TabReD is not a replacement for previous benchmarks, but rather allows to conduct evaluation in different and practically common conditions.
> > > 4. We added a paragraph in conclusion that talks about the intended use of TabReD.
> > > 5. We added a discussion of the distinction of TabReD with time series benchmarks in related work, and also link to appendix where we provided part of our discussion of usefulness of time-series inspired techniques on TabReD.
> > > 6. We added some balance to our discussion of Kaggle competitions. We also changed wording in some places, to avoid being overly positive in respect to Kaggle competitions.
> > >
> > > If you still find that some parts of our paper give off the wrong impression, we are open to modifying the manuscript again!

---

> > > > ### Comment · Reviewer_9YJo · 2024-11-21
> > > > **Score Increase**
> > > >
> > > > Thank you for the updated PDF. My concerns have been fully addressed, and I think the manuscript has been significantly improved. I am raising my score to **8, accept**.

---

### Official Review · Reviewer_8TaW · 2024-11-03

**Soundness:** 4
**Presentation:** 3
**Contribution:** 4
**Rating:** 8
**Confidence:** 3

**Summary:**

This paper identifies and addresses important gaps between academic tabular ML benchmarks and industrial applications. The key contributions are:
* Literature Cleanup: Analysis of 100 existing benchmark datasets, revealing issues like data leakage and underrepresentation of real-world conditions -- primarily (a) lack of temporal drift, and (b) small feature space.
* New Benchmark: Introduction of TabReD, a new benchmark of 8 industry-grade datasets focusing on temporal drift and feature-rich scenarios. Datasets are sourced from Kaggle (4) and proprietary internal company data (4).
* Evaluations: Comprehensive evaluation showing that most recent tabular DL advances do not transfer to these more realistic conditions, and that gradient boosted trees and simple MLPs tend to perform best

**Strengths:**

The paper addresses a critical gap between academic benchmarks and industrial applications of tabular ML, with clear evidence from an analysis of 100 existing academic datasets that prior work is limited by (a) data leakage, (b) lack of temporal splits, (c) containing mostly non-tabular data, such as flattened images.
* Very rigorous empirical methodology:
 * Clear criteria for dataset selection
* Statistical significance testing across 15 random seeds for evaluations
* Comprehensive evaluation of multiple model architectures and techniques

**Weaknesses:**

* No detailed analysis of failure modes for retrieval-based models
   * Possible future extension to important domains — not fully representative (healthcare, science)
   * The authors do explicitly mention this as a specific limitation
* Given the emphasis on industrial applications, it would be nice to have a sense of how impactful these performance shifts are for the downstream tasks, i.e. can you give us a sense of how much 0.01 RMSE matters for cooking time estimation dataset?
* Unclear comparison to prior benchmarks on how your benchmark yields improved model rankings. A central claim of this paper is that prior benchmarks yield model rankings that change once temporal splits are taken into account. Hence, your arguments about XGboost and MLP being surprisingly superior. But you don’t specify how the models that you tested would have been ranked by the prior work benchmarks (e.g. Tabzilla, WildTab, etc.) that lack temporal splits. You get at this a bit in Figure 2 within your own benchmark, but it may help strength the argument to a more detailed comparison between the conclusions of your Table 3 to the rankings that would have been generated by running these models on the prior work instead.

**Questions:**

* How does the subsampling of larger datasets affect the conclusions? Could you provide a sensitivity analysis?
* How did you determine data leakage issues that weren't reported in prior literature?
* Could you elaborate on the feature engineering process for the new datasets and how it reflects industry practices?
* Why did you choose not to include in TabReD the existing datasets where temporal splits are possible due to the presence of temporal information?
* Provide recommendations as to how TabReD should be used with existing benchmarks.

---

> ### Author Response · Authors · 2024-11-20
>
> Thank you for your review! We would like to answer your questions and address the concerns you have about our paper.
>
> > No detailed analysis of failure modes for retrieval-based models
>
> We conduct an additional experiment, where we select top-20 important features using XGBoost feature importance and run several methods on them. Here is the table summarizing average improvement over MLP when using all features or 20 features, in percentages:
>
> | | MLP | MLP [aug, rec] | MLP [ens.] | MLP [plr ens.] | MLP [plr] | TabR | XGBoost |
> |-|-|-|-|-|-|-|-|
> |All Features| 0.0 | -1.98 | 0.67 | 1.28 | 0.61 | -2.78 | 0.91 |
> |Top-20 Features | 0.0 | -0.13 | 0.52 | 0.98 | 0.69 | 1.09 |  0.97 |
>
> \* NOTE: *MLP with all features has an advantage of 5.65% over MLP with 20 features.*
>
> With this change retrieval methods represented by TabR start showing performance gains over MLP. Deeper investigation into this could be an interesting avenue for future work.
>
> > Given the emphasis on industrial applications, it would be nice to have a sense of how impactful these performance shifts are for the downstream tasks, i.e. can you give us a sense of how much 0.01 RMSE matters for cooking time estimation dataset?
>
> On cooking time dataset, 0.01 RMSE would be a pretty significant improvement, potentially deserving of implementation in production pipeline. It would correspond to about 3% improvement of time estimation in minutes.
>
> > it may help strength the argument to a more detailed comparison between the conclusions of your Table 3 to the rankings that would have been generated by running these models on the prior work instead.
>
> We believe that the breadth of comparison in Figure 1 is not a priority. It is addressed by the consistency of our results obtained on the Gorishniy benchmark with the results from the tabular DL literature. For example, retrieval methods (TabR, ModernNCA), embeddings and ensembles show consistent performance across hundreds of datasets in existing benchmarks (the largest one to date is [Han-Jia Ye et al.](https://arxiv.org/abs/2407.00956), it covers many datasets from [Léo Grinsztajn et al.](https://arxiv.org/abs/2207.08815) and Tabzilla). Strenghts of the long-training with augmentations is also replicated on a broader set of datasets ([Kyungeun Lee et al.](https://arxiv.org/abs/2405.07414)).
>
>
> ---
> We would also like to answer your questions.
>
> 1. We conducted experiments on different subsamples. Our conclusions hold quite robustly for different subsampling of the full dataset, i.e. average improvement of XGBoost over MLP is 5.21% ± 0.04% on weather dataset (std is taken over subsamples), and 0.34% ± 0.06% on maps-routing dataset.
> 2. We study the preprocessing pipeline from when the dataset first appeared to its usage in a benchmark, usually a mistake made in this process allows an additional information containing leakage to appear in the data. In some datasets, one of original target variables is added to the features, while some datasets have an answer to one example as a feature in other, and originally used non-random split to avoid leakage, which was changed to random in later literature.
> 3. These datasets are samples from internal data pipelines from an industrial company. We cannot yet reveal exactly how these features were obtained, partly due to anonymity concerns, however we will provide some additional information upon acceptance.
> 4. We also focus on richer feature sets, and most datasets where time feature was available have a small number of features.
> 5. We see two possible use-cases. The first one is to use it to evaluate a method that already shows improved performance on previous benchmarks, to show how well does this method handle a shift to feature-rich and timeshift-aware data. If a model fails to perform on TabReD, one could provide analysis on random splits or 20 feature subsamples we also include in our benchmarks, to better understand the precise reason why a model's performance worsened. The second use-case is for methods that deal specifically with temporal shift or big number of features, since this kind of models can currently be tested only on TabReD.

---

> ### Comment · Reviewer_8TaW · 2024-11-24
> **Thank you!**
>
> Thank you for your response and taking the time to answer my questions!

---

### Official Review · Reviewer_uNfX · 2024-11-04

**Soundness:** 3
**Presentation:** 3
**Contribution:** 3
**Rating:** 8
**Confidence:** 3

**Summary:**

The present paper studies the characteristics of academic tabular benchmarks for deep learning models, and makes the following contributions:

- The authors identify several issues that hinder the reliability of such benchmarks for real-world applications. In particular, they show that 1) time-evolving datasets (where the data distribution changes with time) and 2) feature-rich datasets (e.g. stemming from feature engineering pipelines) are under-represented in academic benchmarks, while they are common in industrial scenarios.

- To address these pitfalls, the authors introduce TabReD, a collection of 8 industry-grade tabular datasets from various domains, and show that they exhibit the 2 aforementioned aspects.

- Finally, they evaluate several tabular models on TabRed. Results seem to show that top-performing architectures in academic benchmarks do not transfer well to TabRed, while simpler models (e.g. GBDT or MLP ensembles) remain performant. The authors attribute this observation to the temporal data drift present in the datasets, and the fact that they have large number of correlated/uninformative features.

**Strengths:**

1) The paper is clear and easy to read
2) The paper has practical implications. Benchmarks are an important aspect of research and limitations of standard ones for tabular learning are demonstrated. The TabReD benchmark can fill this gap by being closer to real-world/industry settings.
3) The claims are overall well supported by experiments and analyses, although several things could be improved (see weakness section):
    - The authors conduct a extensive analysis of 100 datasets used in common tabular learning benchmarks. A significant fraction of them exhibit data quality issues such as leakage, untracability, or a non-tabular format. Many of them are also potentially subject to temporal data drift, but this aspect is overlooked in most benchmarks.
    - The authors provide 8 datasets (including 4 novel ones) from real-world, industrial applications. Figure 3 and 4 confirm that they exhibit temporal data drift, and have large number of features with complex correlation patterns (unlike academic datasets).
    - The benchmark on TabReD includes a wide range of approaches: baselines, top-performing models in academic benchmarks, and training techniques that were also shown to improve accuracy in academic benchmarks. The results seem to show that certain models/techniques which performed better in academic benchmarks actually perform similarly or worse than simple baselines (Fig 1 and Table 3).
    - The authors partly explain these differences with the temporal data drift present in industrial datasets. For this they compare two data splits scenarios (time-based and random, see Figure 2), and show that random splits overestimates model performance (both in mean and std across seeds). They also seem to alter model ranking, although this is not clear.

**Weaknesses:**

1) The average ranks shown in Table 3 are a bit hard to intepret because the methods are often very close in terms of performance, so it is not clear how much difference there is between the worst and best approaches. Figure 1 partly solves this by showing the average percentage of improvement over a MLP baseline, but doesn't show it for all methods. It would be nice to have this information either in Figure 1 or as another column of Table 3 in order to get a better feeling of how the methods perform.
2) Even if recent tabular DL methods seem to perform worse than simple baselines on TabReD, I am not fully convinced that it is due to 1) temporal data drift or 2) datasets with large number of correlated/uninformative features, as the authors imply. Knowing this is important to allow further research to focus on the right aspects when designing novel methods. It would also motivate the choices the authors made when they designed the TabReD benchmark.
    - For instance, Figure 2 compares the performance of a few methods when doing time-based versus random splits of the data. As expected, performance decreases when using time-based splits, but the impact on the models ranking and relative gap is not clear. For instance, TabR seem to perform poorly in both time-based and random splits. In addition, most of the methods considered in Table 3 are not included. I would like to see a table with aggregated results for most methods, such as the average ranking in both splitting scenarios, and the average percentage decrease in performance when going for random to time-based splitting. This will allow to assess the effect of temporal data drift more easily, and see if the baseline models are indeed more resilient to it.
    - The impact of having datasets with large numbers of correlated/uninformative features is not explored. I would like to see a analysis similar to the one done for splitting strategies. For instance, the authors could apply simple filtering techniques to remove highly correlated and/or uninformative features, and compare the average ranks in the "raw" versus "filtered" scenarios, and show the average percentage of improvement per model.

**Questions:**

1. The meaning of needed/possible/used in Table 1 wasn't clear with just the caption. It would be good to have a sentence to explain it in the caption.
2. When calculating model rankings in Table 3, the Appendix (line 902) explains that a model B is ranked better than model A if 1) B is in average better (across 15 random initializations), and 2) the gap in average performance is smaller than the standard deviation of model B performance. First, I guess the authors meant larger instead of smaller? Second, it seems that with this definition the ranking isn't transitive. For example, imagine three models: model A with average R2 score = 0.6 (std = 0.1), model B with average R2 score = 0.4 (std = 0.2) and model C with average R2 score = 0.55 (std = 0.2). In this case model A should be ranked higher than model B and ranked the same as model C. However model C should also be ranked the same as model B, so it is confusing. I maybe misunderstood something here, so I would like to have more details on that.

---

> ### Author Response · Authors · 2024-11-20
>
> Thank you for the review! We would like to address the weaknesses you pointed out in our paper.
>
> > It would be nice to have this information either in Figure 1 or as another column of Table 3 in order to get a better feeling of how the methods perform.
>
> We will add information about the average percentage improvement over MLP in the appendix of the revised paper, shortly.
>
>
> > I am not fully convinced that it is due to 1) temporal data drift or 2) datasets with large number of correlated/uninformative features, as the authors imply.
>
> We provide additional experiments on the effect of large number of features, which helps us demonstrate how both of these points influence a method's performance. We will circle back to this in response to the second point of your second weakness.
>
> > I would like to see a table with aggregated results for most methods
>
> To make the presentation of comparison clearer, we plot relative improvements over MLP when using / not using time splits. Results are in the table below:
>
> |      | MLP (ens.) | MLP (aug. rec.) | MLP-PLR | MLP-PLR (ens.) | TabR    | XGBoost    |
> | ---- | ---------- | --------------- | ------- | -------------- | --- | --- |
> | Temporal Split | 0.67 | -1.98  |  0.61 | 1.28 | -2.78 | 0.91 |
> | Random split | 0.28 | -0.47 | 0.47 | 0.96 | 0.6 | 2.02 |
>
>
> > As expected, performance decreases when using time-based splits, but the impact on the models ranking and relative gap is not clear.
>
> When going from time-split to random splits, the gap between XGBoost and MLP-PLR becomes larger, while the difference between TabR and MLP-PLR becomes smaller.
>
> > In addition, most of the methods considered in Table 3 are not included.
>
> For clearer presentation we chose to focus on one method from each of the major categories: retrieval, embeddings, augmentations. Most other methods claim state-of-the-art results on benchmarks on which they were introduced, which often includes one or several of the ones we observe in related work. For this specific in-depth analysis though, due to time and resource constraints, we chose to focus only on method families and not specific methods.
>
> > The impact of having datasets with large numbers of correlated/uninformative features is not explored.
>
> Thank you for proposing this useful experiment, which helps us demonstrate the point of our paper! We conduct an additional experiment, where we select top-20 important features using XGBoost feature importance and run several methods on them. Here is the table summarizing average improvement over MLP when using all features or 20 features, in percentages:
>
> |  | MLP | MLP [aug. rec] | MLP [ens.] | MLP [plr ens.] | MLP [plr] | TabR | XGBoost |
> |-|-|-|-|-|-|-|-|
> |All Features| 0.0 | -1.98 | 0.67 | 1.28 | 0.61 | -2.78 | 0.91 |
> |Top-20 Features | 0.0 | -0.13 | 0.52 | 0.98 | 0.69 | 1.09 |  0.97 |
>
> \* NOTE: *MLP with all features has an advantage of 5.65% over MLP with 20 features.*
>
> We'll add this result to the PDF shortly. With this change retrieval methods represented by TabR start showing performance gains over MLP. Deeper investigation into this could be an interesting avenue for future work. Furthermore, improved training methodologies for the MLP (aug. rec) are less detrimental in this setup, and they are even helpful for some datasets.
>
> ---
>
> Now, we would like to answer you questions:
>
> 1. We somewhat agree, but we want to avoid causing any misunderstanding with oversimplifying in caption, so we explain what we meant in the main text.
>
> 2. This statistical test follows "Revisiting Deep Learning Models for Tabular Data" for continuation reasons. For more commonly used statistical significance test you can see a multiple comparison test from appendix C.2, Table 5.

---

> > ### Author Response · Authors · 2024-11-25
> > **Invitation to discussion**
> >
> > Dear reviewer, since the discussion period is almost over, we would like to invite you to participate in the discussion.
> >
> > We summarize the new experiments and submission PDF updates made in response to your review:
> >
> > - We changed the presentation of the comparsion between temporal and random splits, and simplified the procedure to compare runs on time-split and randomly split versions of the original TabReD datasets (simpler than prior three sliding window split version). The updated results could be seen in our response to your review and the submission PDF.
> > - We've added additional experimental results exploring the feature rich setting you've suggested to the section 5.4 of the updated PDF
> > - We also highlight the statistical test in Table 6, subsection C.2 in response to your question 2.
> >
> > Do this results and changes address your concerns detailed in "Weaknesses" and "Questions" section? Do you have any remaining questions? We are happy to discuss.

---

> > > ### Comment · Reviewer_uNfX · 2024-11-26
> > >
> > > Thank you for addressing my comments! I am satisfied by the changes made to the paper and updated my rating accordingly.

---

### Author Response · Authors · 2024-11-20
**General Response**

---
## **General Response**

We thank all reviewers for their thoughtfull and helpfull feedback!

We are glad that reviewers (**R-uNfX**, **R-8TaW**, **R-9YJo**) recognize the importance of improving the benchmarking practices in the field of tabular DL and see the role our submission plays in this:
- "The paper has practical implications. Benchmarks are an important aspect of research and limitations of standard ones for tabular learning are demonstrated..." **R-uNfX**
- "The paper addresses a critical gap between academic benchmarks and industrial applications of tabular ML" **R-8TaW**
- "I see this work as eye-opening to many academics and hopefully inspiring better benchmarking practice in the future" **R-9YJo**

We are also glad that **all** reviewers find the experimental results convincing and thorough:
- "The claims are overall well supported by experiments and analyses" **R-uNfX**
- "Very rigorous empirical methodology" **R-8TaW**
- "Given the experimental design and datasets, the results look reasonable and, from the code, reproducible" **R-9YJo**
- "The experiment results are thoroughly presented and explored" **R-s8Za**

---
## **Updates and new experiments**

We summarise the new experiments and discussions inspired by the reviewer's comments below. We plan to incorporate changes into the PDF shortly after discussion with reviewers.

- We explore how the feature-rich aspect of the TabReD datasets affects techniques as suggested by (**R-uNfX**, **R-8TaW**). New results highlight that this is an equally important data characteristic of the new datasets. We'll add the results to the paper, we present the results in **Table 1** below.
- We add more techniques to the comparison in section 5.4 (improved training methodologies and ensembling) as suggested by (**R-uNfX**). This experiment now covers all principally different techniques described in section 5. Results are in the **Table 2** below.
- We provide additional discussion on the TabReD position to answer reviewers **R-9YJo** and **R-s8Za** concerns regarding IID vs non-IID data positioning (see individual responses for the discussion).


**Table 1.** How does extensive feature engineering affects various techniques? Each row represents relative improvement of a method to the MLP. (*MLP with all features has an advantage of 5.65% over MLP with 20 features.*). TabR and MLP [aug, rec] are the most affected methods.

|  | MLP | MLP [aug. rec] | MLP [ens.] | MLP [plr ens.] | MLP [plr] | TabR | XGBoost |
|-|-|-|-|-|-|-|-|
|All Features| 0.0 | -1.98 | 0.67 | 1.28 | 0.61 | -2.78 | 0.91 |
|Top-20 Features | 0.0 | -0.13 | 0.52 | 0.98 | 0.69 | 1.09 |  0.97 |

**Table 2.** A succinct version of the experiment from section 5.4 with additional techniques added (ensembling, aug. rec). The table tells a similar story to the Figure 2, see the difference of the TabR performance and relative difference betwen MLP-PLR and XGBoost in two setups.

|      | MLP (ens.) | MLP (aug. rec.) | MLP-PLR | MLP-PLR (ens.) | TabR    | XGBoost    |
| ---- | ---------- | --------------- | ------- | -------------- | --- | --- |
| Temporal Split | 0.67 | -1.98  |  0.61 | 1.28 | -2.78 | 0.91 |
| Random split | 0.28 | -0.47 | 0.47 | 0.96 | 0.6 | 2.02 |

---

### Meta-Review · Area_Chair_L2qg · 2024-12-12

**Metareview:**

Tabular learning with temporal drift is an interesting problem with many use cases in practice and is hardly discussed or addressed in academic literature. The authors nicely summarise the related work in the paper which is generally well written and easy to understand. The claims made by the authors are supported by comprehensive experiments and analyses (though reviewers pointed out some improvements) on about 100 data sets including four newly introduced ones. In sum a solid piece of work that should be of interest to the community and may encourage more colleagues to study temporal drift in tabular learning.

**Additional Comments On Reviewer Discussion:**

The authors provided more information in the rebuttal that was helpful for the reviewers.

---

### Decision · Program_Chairs · 2025-01-22

Accept (Spotlight)